# Transcription factor DUO1 generated by neo-functionalization is associated with evolution of sperm differentiation in plants

Asuka Higo[1], Tomokazu Kawashima [2,3], Michael Borg[2], Mingmin Zhao[4], Irene López-Vidriero[5], Hidetoshi Sakayama[6], Sean A. Montgomery[2], Hiroyuki Sekimoto [7], Dieter Hackenberg[4], Masaki Shimamura[8], Tomoaki Nishiyama[9], Keiko Sakakibara[10], Yuki Tomita[1], Taisuke Togawa[11], Kan Kunimoto[1], Akihisa Osakabe[2], Yutaka Suzuki[12], Katsuyuki T. Yamato [11], Kimitsune Ishizaki [6], Ryuichi Nishihama [1], Takayuki Kohchi [1], José M. Franco-Zorrilla[5], David Twell[4], Frédéric Berger[2] & Takashi Araki [1]

Evolutionary mechanisms underlying innovation of cell types have remained largely unclear. In multicellular eukaryotes, the evolutionary molecular origin of sperm differentiation is unknown in most lineages. Here, we report that in algal ancestors of land plants, changes in the DNA-binding domain of the ancestor of the MYB transcription factor DUO1 enabled the recognition of a new *cis*-regulatory element. This event led to the differentiation of motile sperm. After neo-functionalization, DUO1 acquired sperm lineage-specific expression in the common ancestor of land plants. Subsequently the downstream network of DUO1 was rewired leading to sperm with distinct morphologies. Conjugating green algae, a sister group of land plants, accumulated mutations in the DNA-binding domain of DUO1 and lost sperm differentiation. Our findings suggest that the emergence of DUO1 was the defining event in the evolution of sperm differentiation and the varied modes of sexual reproduction in the land plant lineage.

---

[1] Graduate School of Biostudies, Kyoto University, Sakyo-ku, Kyoto 606-8501, Japan. [2] Gregor Mendel Institute (GMI), Austrian Academy of Sciences, Vienna Biocenter (VBC), Dr. Bohr Gasse 3, 1030 Vienna, Austria. [3] Department of Plant and Soil Sciences, University of Kentucky, Lexington, KY 40546-0312, USA. [4] Department of Genetics and Genome Biology, University of Leicester, University Road, Leicester LE1 7RH, UK. [5] Unidad de Genómica, Centro Nacional de Biotecnología, CNB-CSIC, Campus de Cantoblanco, C/Darwin 3, 28049 Madrid, Spain. [6] Department of Biology, Graduate School of Science, Kobe University, 1-1 Rokkodai, Nada-ku, Kobe 657-8501, Japan. [7] Department of Chemical and Biological Sciences, Faculty of Science, Japan Women's University, 2-8-1 Mejirodai, Bunkyo-ku, Tokyo 112-8681, Japan. [8] Department of Biology, Graduate School of Science, Hiroshima University, 1-3-1 Kagamiyama, Higashi-Hiroshima 739-8526, Japan. [9] Advanced Science Research Center, Kanazawa University, 13-1 Takara-machi, Kanazawa 920-8640, Japan. [10] Department of Life Science, College of Science, Rikkyo University, 3-34-1 Nishi-Ikebukuro, Toshima-ku, Tokyo 171-8501, Japan. [11] Faculty of Biology-Oriented Science and Technology, Kindai University, Kinokawa 649-6493, Japan. [12] Department of Computational Biology and Medical Sciences, Graduate School of Frontier Sciences, The University of Tokyo, 5-1-5 Kashiwanoha, Kashiwa-shi, Chiba 277-8562, Japan. These authors contributed equally: Asuka Higo, Tomokazu Kawashima. Correspondence and requests for materials should be addressed to F.B. (email: Frederic.Berger@gmi.oeaw.ac.at) or to T.A. (email: taraqui@lif.kyoto-u.ac.jp)

The development of differentiated cell types is related to multicellularity and the acquisition of novel functions. Morphological similarities between cell types across organisms might be indicative of their common evolutionary origin. The common origin of differentiated cell types and their underlying conserved gene regulatory networks illustrate the deep homology[1] that exists amongst distant species[2]. However, while aspects of gene regulatory networks that cause differentiation of specific cell types might be conserved, cells still exhibit distinct traits that evolve independently. Such a scenario is illustrated by the evolution of male gametes. Across eukaryotes, small motile sperm that fertilize large immotile eggs evolved multiple times. This led to a mode of sexual reproduction defined as anisogamy, which is considered a key event in the evolution of sexual dimorphism[3–6]. Across kingdoms, spermatozoa share fusogenic properties[7], yet show a widely varied morphology. In plants, motile sperm evolved first in Charophyceae, the freshwater algae sharing the last common ancestor with land plants, which are collectively referred to as Streptophyta (Fig. 1a). However, sperm motility is absent in conjugating green algae (Zygnematophyceae), the sister group of land plants[8]. Bryophytes, which include liverworts, mosses, and hornworts, are representatives of the first land plants and produce motile sperm that show characteristics identical to sperm from Charophyceae (Fig. 1a). Among vascular plants, while ferns and some gymnosperms produce motile flagellate sperm, flowering plants differentiate non-motile gametes, which are instead transported to the female gametes by the pollen tube. Such distinct modes of spermatogenesis in the land plant lineage suggest either the diversification of an ancestral gene regulatory network or that distinct types of spermatogenesis evolved independently.

In *Arabidopsis thaliana*, DUO POLLEN 1 (DUO1) is a MYB transcription factor (TF) controlling male gamete development and differentiation[9–11]. The network controlled by DUO1 in *Arabidopsis* comprises the two closely related TFs, DUO1-ACTIVATED ZINC FINGER1 (DAZ1) and DAZ2 which, together with DUO1, regulate the expression of the sperm-specific histone variant H3.10 and the fusogenic factors GENERATIVE CELL-SPECIFIC 1 /HAPLESS 2 (GCS1/HAP2) and GAMETE EXPRESSED 2 (GEX2)[12]. Recent access to the transcriptome during male gametogenesis in the model bryophyte *Marchantia polymorpha*[13,14] enabled us to address whether plant sperm share a common molecular origin.

Here we combine genetic and molecular analyses to show that *DUO1*-type MYB TF is present in land plants[15]. Using phylogenetic analyses we further identify *DUO1* orthologs in major clades of bryophytes and in Charophyceae but not in *Mesostigma* and *Klebsormidium* representing earlier diverging groups of freshwater green algae. The ancient origin of DUO1 is supported by conservation of its essential function of DUO1 in spermatogenesis in the land plant lineage. Functional in vivo assays and DNA-binding analyses suggest that this strong conservation is explained by a change in the DNA-binding domain of an ancestral MYB TF in the common ancestor of Charophyceae and land plants. This change enabled DUO1 to bind a distinct motif in its target promoters and led to evolution of a sperm differentiation program. Our results support a single origin for spermatogenesis despite the remarkable diversity of sperm morphology amongst plant species.

## Results and Discussion
### Identification of DUO1 ortholog in *Marchantia*. DUO1 differs from all other R2R3 MYB TFs by a supernumerary lysine residue between the R2 and R3 repeats of its MYB domain[11]. Based on search for this signature amongst MYB TF families in land plants,

DUO1 orthologs were identified in several species of bryophytes, including liverworts and mosses (Fig. 1a, b and Supplementary Figure 1a), indicating a common origin of DUO1 in the ancestor of land plants. In the liverwort *Marchantia polymorpha*, male reproductive branches (antheridiophores) differentiate from a vegetative thallus and host the antheridia where sperm cells are formed[14,16]. In the antheridia, proliferating sperm-cell precursors give rise to spermatid mother cells after a penultimate division. Sperm mother cells then undergo a diagonal cell division to produce spermatids, which differentiate into sperm (Fig. 1c)[16,17]. MpDUO1 is expressed in the sperm-cell lineage, exclusively in spermatid mother cells and spermatids (Fig. 1d, e and Supplementary Figs. 1b–d, 2). The MpDUO1 protein accumulates in the nucleus (Fig. 1f and Supplementary Fig. 1e). In *Haplomitrium mnioides*, a representative of the most basal liverworts, HmnDUO1 is also expressed in antheridia (Supplementary Figure 1f). The Mp*DUO1* promoter confers sperm-cell-lineage-specific expression in *Arabidopsis* mature pollen like the native At*DUO1* promoter (Fig. 1g). Conversely, the At*DUO1* promoter confers an expression pattern similar to that of Mp*DUO1* in developing antheridia (Fig. 1h and Supplementary Figure 1c). These results suggest that transcriptional control of *DUO1* has remained conserved between liverworts and angiosperms, which diverged more than 450 MYA[13].

In *Arabidopsis*, *duo1* null mutants fail to undergo G2-to-M transition of the generative cell followed by differentiation of two sperm cells[11]. *Marchantia polymorpha* is dioecious with separate sexes determined by sex chromosomes. *Marchantia* male and female plants carrying a null Mp*duo1-1*$^{ko}$ allele (Supplementary Figs. 1b, 3a) show normal vegetative development. Female Mp*duo1-1*$^{ko}$ plants are fertile, but males Mp*duo1-1*$^{ko}$ are sterile (Fig. 2a, b and Supplementary Figure 3b), suggesting a specific role of MpDUO1 in male gametogenesis. The discharge of sperm masses as white aggregates from antheridiophores upon hydration is observed in wild-type (WT) but not in Mp*duo1-1*$^{ko}$ (Fig. 2b). In Mp*duo1-1*$^{ko}$, the final diagonal division that produces spermatids takes place as in WT, but the subsequent sperm differentiation fails (Fig. 2c–e and Supplementary Figure 3c, d). In WT spermiogenesis, the sperm nucleus condenses and acquires a crescent shape that elongates as sperm mature. During formation of two flagella and nuclear elongation, microtubule arrays assemble the spline, which serves as a backbone structure for the elongated nucleus and motility apparatus located at the base of flagella. Eventually, reduction of cytoplasm occurs, and the mature sperm become almost devoid of cytoplasm. In Mp*duo1-1*$^{ko}$, reduction of cytoplasm occurs as in WT, but neither nuclear condensation nor elongation takes place, producing round nuclei in sperm that eventually die (Fig. 2c, d and Supplementary Figures 3c, d). Neither the spline nor flagella are formed (Fig. 2d, e and Supplementary Figure 3d). All defects observed in Mp*duo1-1*$^{ko}$ sperm can be complemented by the expression of wild-type MpDUO1 (MpComp; Figs. 2c, e, 3a and Supplementary Figures 3a, c, e), showing that MpDUO1 is essential for male gamete development in *Marchantia*. The expression of AtDUO1 under control of the Mp*DUO1* promoter results in partial complementation of the Mp*duo1-1*$^{ko}$ phenotype (AtComp; Figs. 2c, 3a and Supplementary Figures 3a, c, e). We also tested whether MpDUO1 can substitute for AtDUO1 in *Arabidopsis*. In *Arabidopsis duo1* mutants (At*duo1*), the sperm precursor fails to divide, resulting in a single infertile cell that does not express the male germline-specific histone H3 variant (H3.10) encoded by *HISTONE THREE RELATED 10* (*HTR10*)[9–11,18]. The expression of MpDUO1 under control of the At*DUO1* promoter almost completely rescues the division defect of At*duo1* pollen[11] (MpComp; Fig. 3b). The seemingly rescued sperm do not express the AtDUO1 target gene *HTR10* and are unable to transmit the

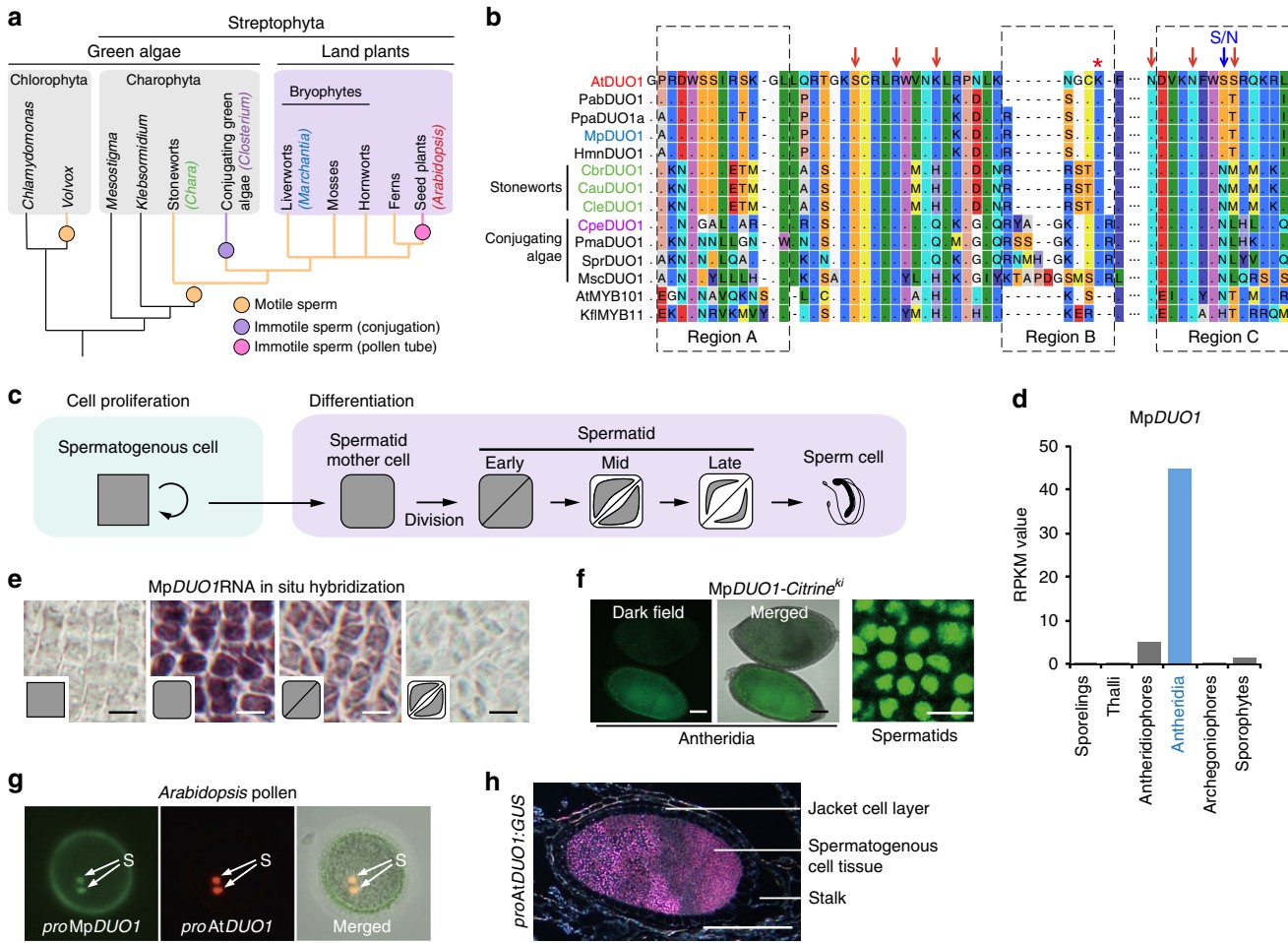

**Fig. 1** Characterization of MpDUO1. **a** A schematic phylogeny indicating the major groups discussed in this study. **b** Amino acid sequence alignment of the DUO1 MYB domains from streptophytes together with the non-DUO1 MYB domains from AtMYB101 and *Klebsormidium* KflMYB11. Regions A, B, and C are highlighted with a dashed box. Red arrows point to putative DNA-interacting amino acid residues. The blue arrow indicates the N to S change that occurred before land plant diversification. The red asterisk indicates the additional K residue typical of the DUO1 subfamily. **c** Schematic view of *Marchantia* male gamete development. **d** MpDUO1 expression profile during *Marchantia* sexual life cycle based on RNA-seq data of major tissues analyzed in ref. [14] Antheridiophores and archegoniophores are male and female reproductive branches, respectively. Thalli contain haploid somatic vegetative tissue. Sporophytes are diploid and develop from the embryo after fertilization. **e** RNA in situ hybridization of MpDUO1 during male gamete development (see Supplementary Figure 2 for the sense-probe control). Insets represent the developmental stages referred to in **c**. **f** The expression pattern of MpDUO1-Citrine fusion protein (green) in antheridia containing developing sperm (left two panels) and differentiating spermatids (right panel) from MpDUO1-Citrine[ki] plants. **g** Z-projected confocal images of *Arabidopsis* pollen expressing *pro*MpDUO1:H2B-Clover (green), and *pro*AtDUO:H2B-mRuby2 (red). The pollen grain wall displays autofluorescence in the green channel. S, sperm cells. **h** Dark-field image of a section of an antheridium from a GUS-stained (pink) *Marchantia* plant expressing *pro*AtDUO1:GUS. Scale bars, 5 μm (**e**), 100 μm (**f**, antheridia), 5 μm (**f**, spermatids), and 100 μm (**h**)

At*duo1* allele to progeny. But sperm differentiation is fully restored when the MpDUO1 DNA-binding domain is fused to the AtDUO1 C-terminal activation domain and the resultant chimera is expressed under control of the At*DUO1* promoter (Chimera 1; Fig. 3b). Similarly, expression of the chimera combining AtDUO1 DNA-binding domain with the MpDUO1 C-terminal activation domain results in a better rescue of the Mp*duo1* phenotype (AtDUO1/MpDUO1 chimera; Fig. 3a and Supplementary Figure 3e). Together these results show that DUO1 orthologs have controlled sperm-cell differentiation and morphogenesis since the evolution of land plants.

**Molecular changes leading to DUO1 function.** To dissect the distinct properties of the DUO1 DNA-binding domain, we tested the *trans*-activation potential of DUO1 TFs as well as chimeras thereof using an in vivo luciferase reporter assay with the DUO1-responsive *HTR10* promoter[9]. In contrast to MpDUO1 (Chimera

1), the closely related MYB transcription factor MpR2R3-MYB21 (Fig. 4a and Supplementary Figure 4) does not *trans*-activate *HTR10* (Chimera 2; Fig. 4b), so we used its R2R3 MYB domain as a negative control. Based on sequence alignment, we identified three specific conserved regions (A, B, and C) in the DUO1 DNA-binding domain (Figs. 1b, 4a). Region B includes the super-numerary lysine specific to DUO1 and region C contains several putative DNA-interacting residues, including a serine absent from MpR2R3-MYB21. We generated chimeric constructs swapping these three specific regions between MpDUO1 and MpR2R3-MYB21 and compared their *trans*-activation potential (Chimera 3 to 7; Fig. 4b). The exchange of region B or C significantly reduced activity, suggesting that these regions determine binding specificity for a distinct *cis*-regulatory element. Further, a screen for preferentially-bound DNA motifs using a protein-binding DNA microarray[19,20] demonstrated the requirement of regions B and C for recognition of the conserved DUO1 consensus motif (Fig. 4c)

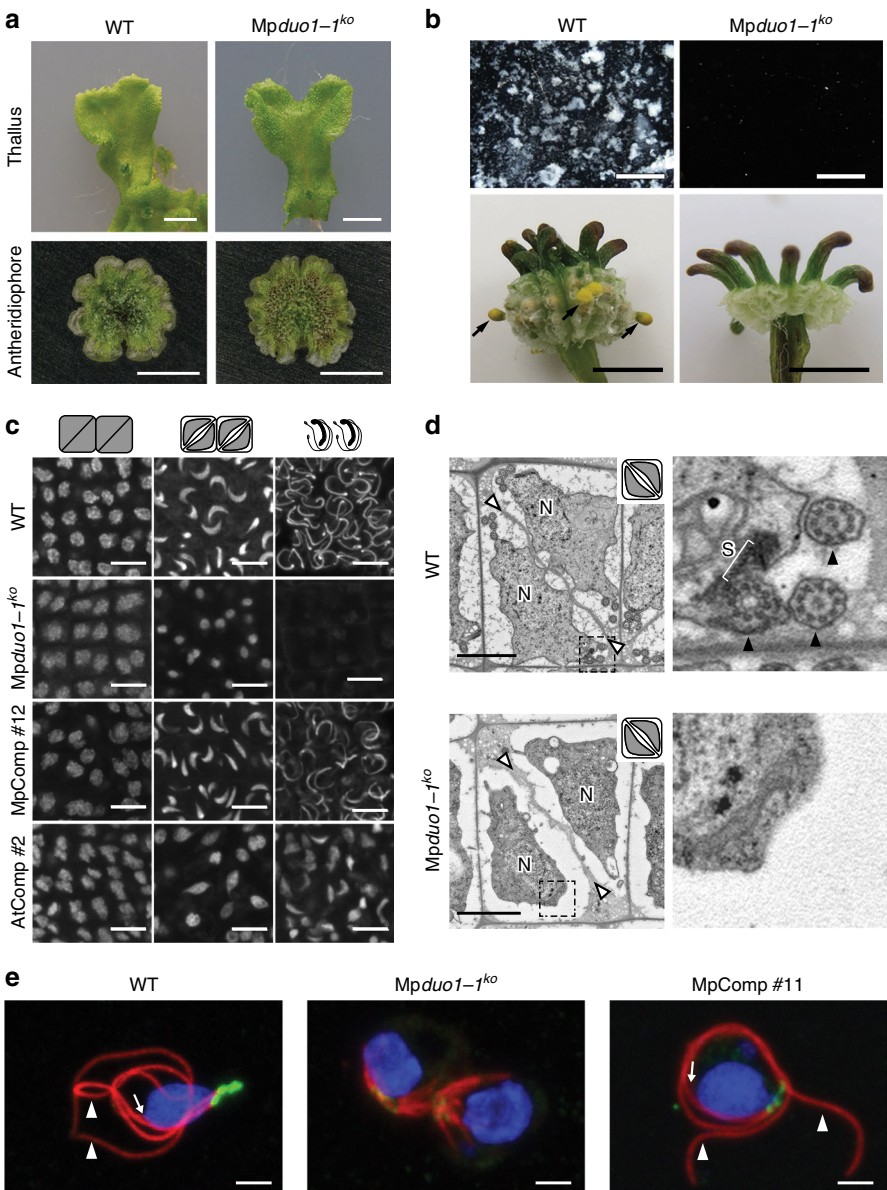

**Fig. 2** Mp*duo1-1*[ko] shows defects in sperm morphogenesis. **a** Images of thallus (top) and antheridiophore (bottom) of WT (left) and Mp*duo1-1*[ko] (right). **b** Top panels show sperm masses (white) discharged from mature antheridiophores into water in WT but not in Mp*duo1-1*[ko]. Bottom panels show the production of yellow sporangia (black arrows), which mark successful fertilization, on female WT about a month after crossing with male WT, but no sporangia production in crosses with male Mp*duo1-1*[ko]. **c** Feulgen staining of spermatids during spermiogenesis in WT, Mp*duo1-1*[ko], and Mp*duo1-1*[ko] complemented by Mp*DUO1* genomic fragment (MpComp) and by *pro*Mp*DUO1*:At*DUO1* (AtComp). **d** Transmission electron micrographs of spermatids. White arrowheads indicate cell boundaries and *N* indicates nucleus. Areas in the dashed line boxes are enlarged in the right panels and show that the spline (S), a microtubular backbone-like structure, and flagella (black arrowheads) are present in WT but missing in Mp*duo1-1*[ko]. **e** Localization of centrins (green) and tubulins (red) in differentiating spermatids of WT, Mp*duo1-1*[ko] (the panel shows two sister cells), and MpComp. Centrin signals mark two basal bodies and an associated multi-layered structure. The blue signals indicate the DAPI stained nuclei. White arrowheads and arrows indicate flagella and spline, respectively. Scale bars, 5 mm (**a**), 400 μm (**b**, top), 5 mm (**b**, bottom), 5 μm (**c**), and 2 μm (**d**, **e**)

distinct from that of other R2R3 MYB TFs[19,20]. Modeling the structure of MpDUO1 in complex with DNA shows that region B is located at the junction between two DNA-interacting helices (Fig. 4a), suggesting that variations at this position might alter DNA-binding specificity due to conformational changes. The region C contacts DNA and it is likely that discrete changes in this region consolidated the specific DNA-binding properties of DUO1. Thus, mutations in regions B and C conferred ancestral DUO1 with the potential to activate transcription of a new set of target genes.

**Impact of MpDUO1 on gene expression in differentiating sperm.** Next, we studied the impact of MpDUO1 on the expression of a set of genes associated with sperm differentiation (Supplementary Figure 5a). From the *Marchantia* antheridium transcriptome[14], we selected genes encoding proteins likely to be important for flagella formation. These included the antheridium-specific tubulin genes Mp*TUA5* and Mp*TUB4*[21], Mp*PACRG*[22], the ortholog of the gene *PARKIN COREGULATED* (*PACRG*) which is associated with the axoneme and is essential for normal sperm formation in mammals[23–25], a homolog of the gene

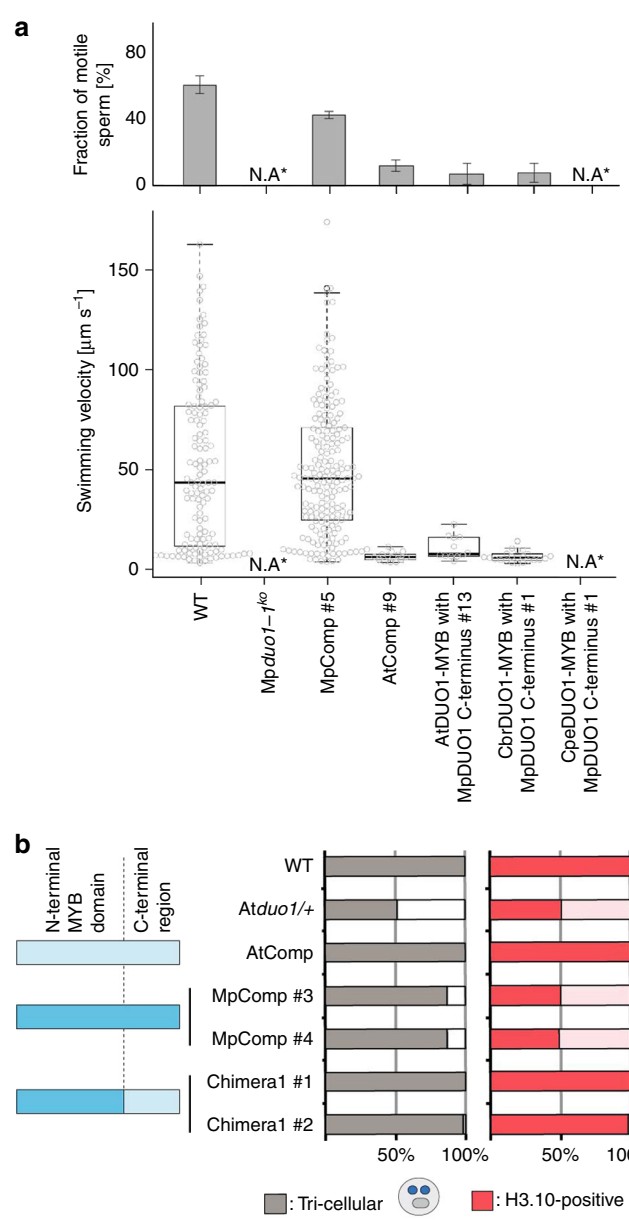

**Fig. 3** Functional conservation of DUO1 MYB domains. **a** Motility of mature sperm cells of WT and Mp*duo1-1^ko* complemented with the Mp*DUO1* genomic fragment (MpComp #5), *pro*Mp*DUO1*:At*DUO1* (AtComp #9), *pro*Mp*DUO1*:At*Chimera* (AtDUO1-MYB with MpDUO1 C-terminus #13), *pro*Mp*DUO1*:Cbr*Chimera* (CbrDUO1-MYB with MpDUO1 C-terminus #1), and *pro*Mp*DUO1*:Cpe*Chimera* (CpeDUO1-MYB with MpDUO1 C-terminus #1). Percentage of motile sperm (upper) and average velocity (lower) of motile sperm based on analysis of three movies for each line are shown. N. A., not applicable, because spermatids abort in Mp*duo1-1^ko* and Mp*duo1-1^ko* complemented with *pro*Mp*DUO1*:Cpe*Chimera* (see Fig. 2b and Supplementary Figure 3e). The numbers of sperm observed are: n = 25, 23, 155 (WT), n = 46, 51, 58 (MpComp #5), n = 27, 32, 25 (AtComp #9), n = 7, 80, 94 (AtChimera #13), n = 44, 50, 59 (CbrChimera #1) for analysis of fraction of motile sperm; n = 117 (WT), n = 67 (MpComp #5), n = 10 (AtComp #9), n = 8 (AtChimera #13), n = 12 (CbrChimera #1) for analysis of swimming velocity. **b** Genetic complementation of At*duo1* by At*DUO1*, Mp*DUO1*, a chimera of the DNA-binding domain of Mp*DUO1* with the activation domain of At*DUO1*. The degree of complementation is measured by rescue of pollen phenotype (left) and expression of H3.10 encoded by *HTR10* (right). Schematic diagram of the constructs used for complementation are shown on the far left. Construct parts are color-coded light blue (AtDUO1) and dark blue (MpDUO1) (see Fig. 4b for details). n > 200 pollen

*DYNEIN LIGHT CHAIN 7* (LC7)[26] (Mp*LC7*[14]), and Mp*CEN1*, a homolog of the gene encoding CENTRIN, which is essential for motility apparatus formation in *Marsilea*[27]. The gene *PROTAMINE-LIKE* (Mp*PRM*)[14] which encodes a protamine-like arginine-rich protein presumably involved in chromatin compaction and nuclear morphogenesis was also included. The expression of most of those genes followed Mp*DUO1* expression (Fig. 5a and Supplementary Figure 2; for Mp*LC7*, Mp*PACRG*, and Mp*PRM*, see ref. [14]) and depended on Mp*DUO1* (Fig. 5b and Supplementary Figure 5b), illustrating how Mp*DUO1* is essential for several key features of sperm differentiation. Moreover, Mp*DUO1* controls expression of Mp*DAZ1*, the *Marchantia* ortholog of At*DAZ1* and At*DAZ2*, two closely related downstream transcription factors targeted by DUO1 in *Arabidopsis*[9,12] (Fig. 5, Supplementary Figs. 2, 5, and Supplementary Table 1), suggesting conservation of the *DUO1-DAZ1* module among land plants. In contrast, Mp*DUO1* does not control expression of orthologs of AtDUO1 targets At*GCS1/HAP2* and At*GEX2*, which are involved in gamete fusion and attachment, respectively[9,28–30,]

and expressed in differentiating *Marchantia* sperm (Fig. 5b, Supplementary Figure 5, and Supplementary Table 1). This independence might be related to the fact that *GCS1/HAP2* and *GEX2* evolved prior to *DUO1* in ancestors of plants, algae, and animals[14,28,29,31]. In *Arabidopsis*, *HTR10* encoding a male germline-specific histone H3 variant (H3.10) is an important target of DUO1[9]. But, no corresponding male germline-specific histone H3 variant was found in *Marchantia*[14]. Instead, a gene encoding a protamine-like sperm nuclear protein (Mp*PRM*)[14] is under the control of MpDUO1 (Fig. 5b and Supplementary Figure 5b). These data suggest that while the *DUO1-DAZ1* regulatory module is conserved, loss and recruitment of new downstream target genes took place after the emergence of vascular plants.

**Lack of impact of RWP-RK transcription factors in sperm**. The RWP-RK domain containing transcription factors Mp*RKD* and Mp*MID* (also known as Mp*RWP2*; Fig. 6a) are also expressed during male gametogenesis[14,32,33]. Mp*RKD* expression is not confined to antheridium development[14,32,33] (Supplementary Figure 5a). In antheridia, Mp*RKD* expression occurs earlier than Mp*DUO1* expression and does not require Mp*DUO1* (Fig. 5 and Supplementary Figs. 2, 5b). In Mp*rkd-4^ge* mutant, some sperm progenitors express Mp*DUO1* (Fig. 6b and Supplementary Figure 6a) and generate fertile sperm[32]. Altogether, these results suggest that Mp*RKD* plays a minor role in spermatogenesis compared with Mp*DUO1*. Mp*MID* is co-expressed with Mp*DUO1* (Fig. 5a and Supplementary Figs. 2, 5a), but Mp*DUO1* is not required for Mp*MID* expression (Fig. 5b and Supplementary Figure 5b). Conversely, the loss of Mp*MID* does not affect the expression of Mp*DUO1* or other sperm-cell-specific genes and, furthermore, male Mp*mid-1^ko* mutants are fertile (Fig. 6c, d, Supplementary Figure 6b, c, and Supplementary Movies 1, 2). A *MID* ortholog is absent from *Arabidopsis*[32–34]. Taken together, these suggest that *MID* does not have a major role in male gametogenesis in land plants, in sharp contrast to its central role in sperm differentiation in the multicellular green alga *Volvox carteri*[35,36], which belongs to distant relatives of land plants (Chlorophyta, Fig. 1a).

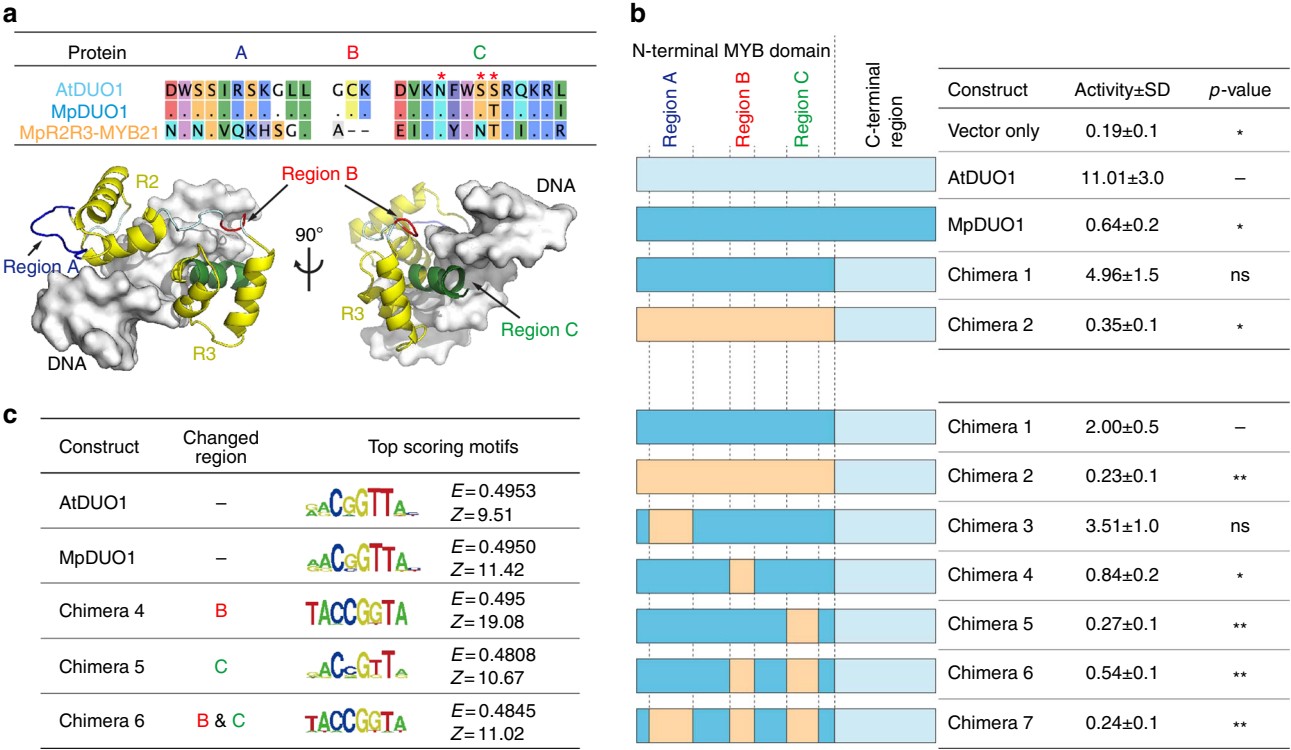

**Fig. 4** Characterization of DUO1 transcription factor activity. **a** Amino acid sequence alignment of regions A, B, and C among AtDUO1, MpDUO1, and MpR2R3-MYB21 (top). Dots indicate matching residues with AtDUO1. Asterisks indicate putative DNA-interacting residues in region C. Structural modeling of MpDUO1 in complex with DNA using SWISS-MODEL (bottom). The MpDUO1 MYB domain is overlaid onto the structure of the AMV v-MYB-DNA complex (PDB code: 1H8A). **b** in vivo transcriptional activation potentials of MpDUO1 and chimeras. Schematic diagram of constructs (left) are color-coded light blue (AtDUO1), dark blue (MpDUO1), and orange (MpR2R3-MYB21). DUO1 transcriptional activation potentials were measured by relative luciferase activity (right). $n = 4$ (upper), $n = 8$ (lower) (**$p < 0.01$; *$p < 0.05$; ns not significant; Student's $t$-test). **c** Position weight matrix representations of the top-scoring 8-mer DNA sequences bound by different MYB domains on a protein-binding DNA microarray. Numbers denoted on the right side of each motif represent the motif E- and Z-scores[19]

**Origin of DUO1 in freshwater green algae**. To further explore the origin of *DUO1*, we extended our search for *DUO1* orthologs in freshwater green algae, the sister groups of land plants (Fig. 1a). We did not find *DUO1* orthologs in the most basal algae *Mesostigma* and *Klebsormidium*, which do not differentiate sperm[37]. In contrast, stoneworts (Charophyceae) produce motile bi-flagellate sperm similar to those of liverworts[37]. In stoneworts, we did not find the ortholog of MID but discovered that they express the *DUO1* ortholog (Fig. 1b) with a pattern broader than the sperm lineage-specific expression in land plants (Fig. 7a and Supplementary Figure 7). CbrDUO1 from the stonewort *Chara braunii* binds to the same motif as other DUO1 proteins, *trans*-activates the DUO1 target *HTR10 in planta* (CbrChimera; Fig. 7b, c), and complements the *Marchantia* Mp*duo1-1$^{ko}$* mutant as efficiently as At*DUO1* (AtDUO1/MpDUO1 and CbrDUO1/MpDUO1 chimeras; Figs. 3a, 7d and Supplementary Figure 3e). These results suggest that an ancestral *DUO1* gene first evolved in a stonewort ancestor and that DUO1 maintained its DNA-binding specificity and function during evolution leading to the land plants. It will be of particular interest to know whether an ortholog of DUO1 exists in *Coleochate*, which belongs to Charophyte and produces motile bi-flagellate sperm similar to those of liverworts[37]. But this lack of identification may be the result of limited availability of genome and transcriptome to vegetative cells only.

Sperm and egg differentiation were lost and replaced by differentiation of mating types in conjugating green algae (Zygnematophyceae) (Fig. 1a). In contrast to stoneworts and land plants, *DUO1* orthologs in conjugating green algae have

accumulated insertions and substitutions in the DNA-binding domain (Fig. 1b, Supplementary Figure 4, and Supplementary Table 2). Consistent with the loss of sperm differentiation in *Closterium peracerosum-strigosum-littorale* complex, CpeDUO1 does not bind to the same motif as other DUO1 orthologs in vitro, does not efficiently *trans*-activate the DUO1 target *HTR10 in planta* (CpeChimera; Fig. 7b, c), and fails to complement the *Marchantia* Mp*duo1-1$^{ko}$* mutant (CpeDUO1/MpDUO1 chimera; Fig. 3a and Supplementary Figure 3e). In addition, CpeDUO1 is barely expressed in gametes (Fig. 7e). Altogether, these data suggest that *DUO1* function is not conserved in conjugating green algae, consistent with the notion that DUO1 was essential for sperm differentiation in algal ancestors of land plants.

In conclusion, we show that *DUO1* orthologs are essential for sperm differentiation in the land plant lineage. This supports the evolution of DUO1-type MYB TFs as a major event leading to the emergence and maintenance of sperm differentiation in the land plant lineage. We propose that a key change in region B took place in the common ancestor of stoneworts and land plants, defining the distinct DNA sequence-specificity and function of the ancestral DUO1 (Fig. 8). The change in region B event was followed by positive selection for *cis*-elements in the *DUO1* promoter that led to sperm lineage-specific expression in the common ancestor of land plants. As land plants evolved, *DUO1* retained a central control over this differentiation program, with gradual reconfiguration of downstream target genes, leading to sperm with distinct cytological features like bi-flagellate motile sperm in bryophytes and non-flagellate immotile sperm delivered

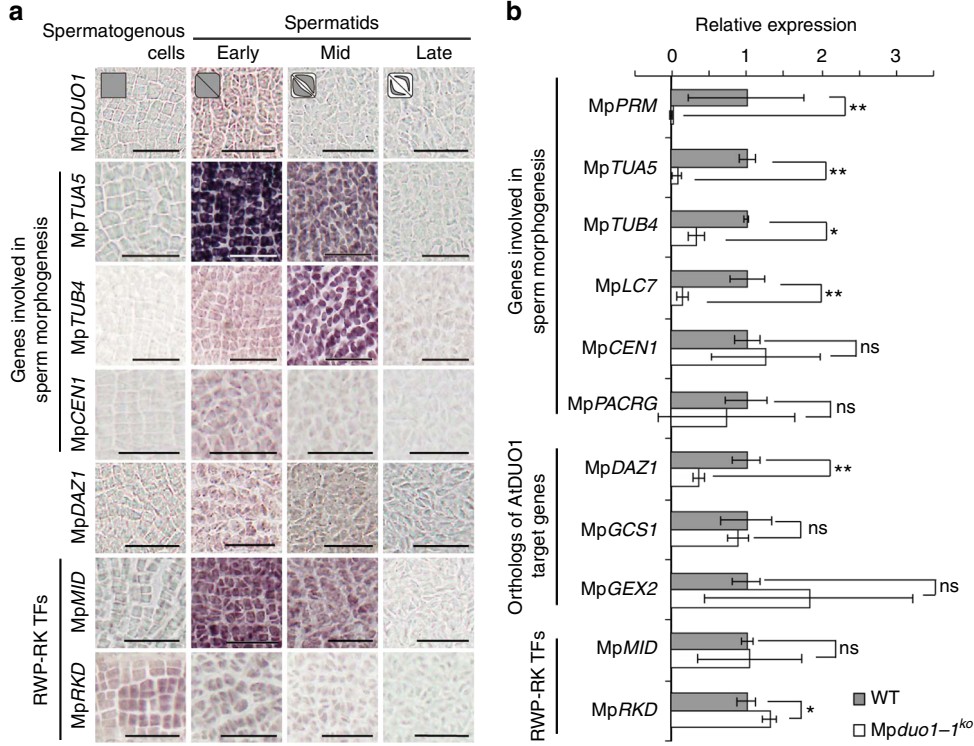

**Fig. 5** Characterization of potential DUO1 target genes in *Marchantia*. **a** RNA in situ hybridization of genes involved in sperm morphogenesis, the ortholog of AtDAZ1 and AtDAZ2 genes, and RWP-RK transcription factors during spermatogenesis (see Supplementary Figure 2 for sense-probe controls). Bars, 25 μm. Cell shape schematics (top) represent the developmental stages referred to in Fig. 1c. **b** Expression levels of genes involved in sperm morphogenesis, the ortholog of AtDUO1 target genes, and RWP-RK transcription factors in antheridiophores of WT (gray bars) and Mp*duo1-1^ko* (white bars). The expression of each gene in WT is set to 1. Error bars indicate mean ± SD; *n* = 3 (**p* < 0.05; ***p* < 0.01; ns, not significant; Student's *t*-test)

by pollen tube in angiosperms. It will be of interest to determine if a comparable parsimonious mode of evolution took place in animals, which also show remarkable diversity of sperm morphologies.

## Methods

**Plant materials and growth conditions**. Male and female accessions of *Marchantia polymorpha* L., Takaragaike (Tak)-1 and Tak-2, respectively, were used as wild-type plants[38]. Plants were cultured on half-strength Gamborg's B5 medium containing 1% sucrose and 1.3% agar under continuous light condition from white fluorescent tubes (50 to 60 μmol m$^{-2}$ s$^{-1}$) at 22 °C. To induce the sexual reproduction, thalli developed from gemmae on half-strength Gamborg's B5 medium were transferred to soil under continuous white light supplemented with far-red light irradiation[39]. *Arabidopsis thaliana* plants were grown on soil in growth chambers at 24 °C under continuous illumination. *duo1-4/+* has been described previously[12]. Male plants of *Haplomitrium minioides* were collected in Higashi-Hiroshima City (Hiroshima Prefecture, Japan) on 24 May, 2012. Plants of *Chara braunii*, *C. australis*, and *C. leptospora* (=*C. globulosa*[40,41]) that have been maintained in culture in Sakayama lab (Kobe University, Japan)[41] were used. The strains of heterothallic *Closterium peracerosum-strigosum-littorale* complex used in this work were NIES-67 (mt$^+$) and NIES-68 (mt$^-$), which were obtained from the National Institute for Environmental Studies, Ibaraki, Japan. The respective vegetative cells were cultured in nitrogen-supplemented medium (C medium; http://mcc.nies.go.jp/02medium-e.html#c) under a 16 h light/8 h dark cycle.

**Plasmid constructions and plant transformation**. The genome regions of Mp*DUO1* and Mp*MID* used for plasmid constructions in this study are shown in Supplementary Figs. 1b and 6b, respectively. To generate the Mp*duo1-1^ko* and Mp*mid-1^ko*, the upper and lower arms were amplified from Tak-1 genomic DNA and were cloned into the *Pac*I and *Asc*I sites, respectively, of the pJHY-TMp1 vector[42]. To generate the Mp*DUO1-Citrine^ki*, the upper and lower arms were amplified from Tak-1 genomic DNA and were cloned into the *Asc*I and *Pac*I sites, respectively, of pJHY-TMp1-Cit in which a Citrine ORF was cloned into the *Hin*dIII site of pJHY-TMp1 vector. The resultant plasmids were introduced into sporelings derived from crosses between Tak-1 and Tak-2. Screening for gene-targeted lines was performed by genomic PCR[42]. A single gemma from each T1 line was isolated to establish the line.

To generate Mp*rkd* mutants in Mp*DUO1-Citrine^ki* background, *RKD* locus was edited using the CRISPR-Cas9 system[43]. Candidates were selected on the basis of gemma cup phenotype[32,33] and PCR products with genomic DNA as the template were analyzed for each candidate to confirm mutations. Several male mutant lines with phenotypes in antheridia[32] were chosen for analysis of Mp*DUO1-Citrine* expression.

To construct *pro*Mp*DUO1:GUS*, the genomic fragment of the upstream region of Mp*DUO1* (proMpDUO1-1) was amplified from Tak-1 genomic DNA and cloned into the *Eco*RI site of pENTR1A vector (Life Technologies) and then transferred to pMpGWB304[44] through the Gateway technology (Life Technologies). The resultant plasmid was introduced into regenerated thalli of Tak-1 and Tak-2. To construct *pro*Mp*DUO1:AtDUO1*, the 5′-upstream region (proMpDUO1-2) and 3′-downstream region (MpDUO1 3′ region) of Mp*DUO1* were amplified from Tak-1 genomic DNA. The proMpDUO1-2 was cloned between *Sal*I and *Bam*HI sites and MpDUO1 3′ region was cloned in *Eco*RV site, respectively, of pENTR1A to generate pENTR1A_proMpDUO1_3′ MpDUO1. AtDUO1 ORF was inserted between *Bam*HI and *Not*I sites of pENTR1A_proMpDUO1_3′ MpDUO1 and the *pro*Mp*DUO1:AtDUO1:3′* MpDUO1 fragment was transferred into pMpGWB301[44] through the Gateway technology. To construct *pro*Mp*DUO1:MpDUO1* for MpDUO1-1^ko complementation, the genomic fragment containing the 5′-upstream region and coding region of Mp*DUO1* was amplified from Tak-1 genomic DNA. The fragment was cloned into pDONR221 (Life Technologies), then into pMpGWB101[44] through the Gateway technology (Life Technologies). The resultant plasmids were introduced into regenerated thalli of Mp*duo1-1^ko*. T1 lines were selected on the half-strength Gamborg's B5 medium containing 0.5 μM chlorsulfuron and a single gemma from each T1 line was isolated to establish the line.

For *Arabidopsis* complementation assay in At*duo1-4/+*, *pro*At*DUO1:AtDUO1-Clover*, *pro*At*DUO1:MpDUO1-Clover* and *pro*At*DUO1:Chimera1* were inserted into the multisite Gateway T-DNA destination vector pAlligatorR43[45] through the Gateway technology (Life Technologies). AtDUO1 cDNA synthesized from Col-0 pollen RNA as well as MpDUO1 cDNA synthesized from Tak-1 young antheridiophore RNA were cloned into pDONR221 with *att*B1 and *att*B2 sites. The additional Gateway technology vectors used in this study are pENTR_2-r3_Clover[45], pENTR_4-1r_proAtDUO1[9,10], and pAlligatorG43_*proHTR10:HTR10-mCherry*[45].

For promoter conservation assay, *pro*Mp*DUO1:H2B-Clover* and *pro*At*DUO1:H2B-Clover* were inserted into the multisite Gateway T-DNA destination vector pAlligatorR43 and pAlligatorG43[45], respectively. Promoters of Mp*DUO1* (proMpDUO1-2) and At*DUO1* were cloned into pDONR-P4P1r (Life

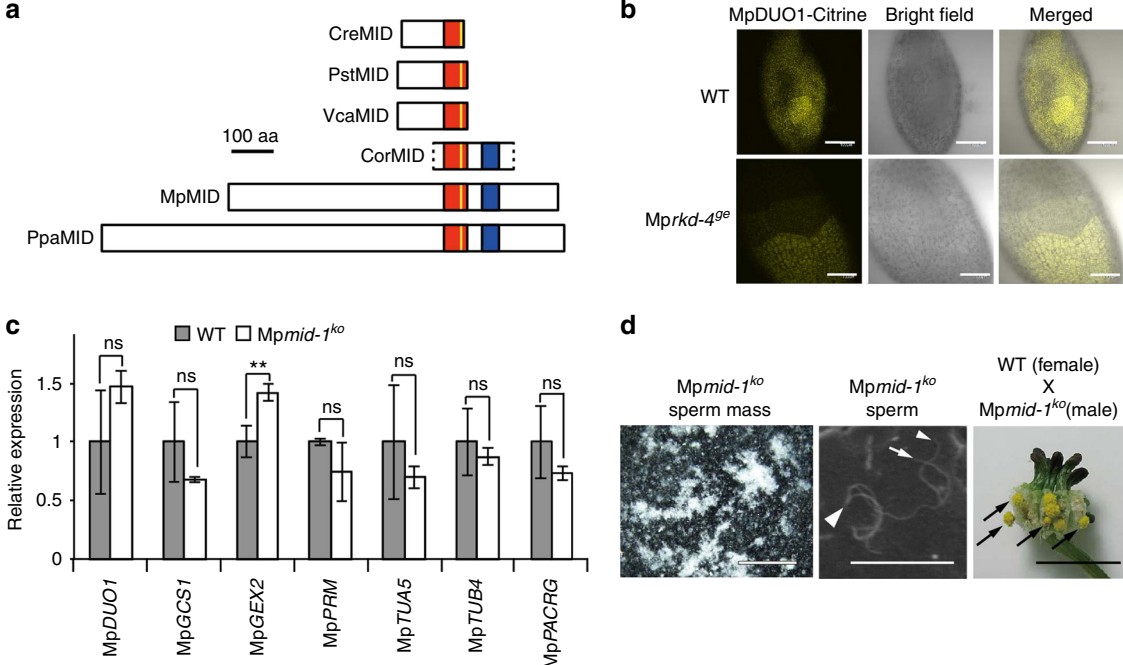

**Fig. 6** Characterization of two RWP-RK transcription factor genes, Mp*MID* and Mp*RKD*, in *Marchantia*. **a** Schematic diagrams of *Volvocine* MID proteins and proteins of MID/RKD(C) subfamily[33] from Streptophytes. The red box represents the conserved RWP-RK domain while the yellow line indicates the RWP-RK motif. A blue box represents a conserved domain outside of the RWP-RK domain[33], which corresponds to motif #12 of ref. [34] CreMID: *Chlamydomonas reinhardtii* (DQ355812), PstMID: *Pleodorina starrii* (BAF42661), VcaMID: *Volvox carteri* (ADI46915), CorMID: *Coleochaete orbicularis* (GBSL01000368 [partial sequence]), MpMID: *Marchantia polymorpha* (KU987912), PpaMID: *Physcomitrella patens* (XM_001779010). MID orthologs are not found in *Arabidopsis thaliana* and in *Chara braunii*. **b** The expression pattern of MpDUO1-Citrine fusion protein in developing antheridia of Mp*DUO1-Citrine*[ki] (WT) and Mp*DUO1-Citrine*[ki]; Mp*rkd-4*[ge] (Mp*rkd-4*[ge]) plants. Note that some sectors of spermatogenous cells differentiate to sperm while others fail in Mp*rkd* mutant antheridia[32]. **c** Quantitative real-time PCR analysis of Mp*DUO1*, Mp*GCS1*, Mp*GEX2*, and genes involved in sperm morphogenesis in stage 4 antheridiophores of WT (gray bars) and Mp*mid-1*[ko] (white bars). The relative expression of each gene in Mp*mid-1*[ko] was compared with WT. Mp*ACT1* was used as an internal control. Error bars indicate mean ± SD; $n = 3$. (\*\*$p < 0.01$; ns, not significant; Student's *t*-test). **d** The sperm masses (white) discharged from mature antheridiophores of Mp*mid-1*[ko] into water (left) and the scanning electron micrograph of sperm of Mp*mid-1*[ko] (middle). White arrowhead and arrows indicate the nucleus and flagella, respectively. The right panel shows mature sporangia (black arrows) on a WT archegoniophore a month after crossing with sperm from Mp*mid-1*[ko]. Scale bars, 100 μm (**b**), 500 μm (**d**, left), 15 μm (**d**, middle), and 5 mm (**d**, right)

Technologies). pENTR221-H2B vector was previously described[45]. The mRuby2 clone (Addgene) was introduced into pDONR-P2rP3 (Life Technologies).

For transient luciferase assay, the firefly and *Renilla* luciferase constructs built with pK7m24GW3[46] and pB2GW7[47], respectively, were used[9]. Chimera 1 to 7 DNA fragments with attB1 and attB2 sites were generated by fusion PCR method[48], and then cloned into pDONR221 vector (Life Technologies) through the Gateway technology. These entry clones as well as At *DUO1* and Mp *DUO1* cDNA entry clones were then recombined into pB2GW7 to generate Pro35S constructs for the transient luciferase assay.

For the protein-binding DNA microarray assay, synthesized *Escherichia coli*-codon-optimized fragments (ThermoFisher Scientific) encoding MYB domains of AtDUO1, MpDUO1, Chimera 4, 5, 6 as well as MpR2R3-MYB21 and KflMYB were cloned into pDONR221 and then transferred to the destination vector pMAL-C2 vector (New England Biolabs) through Gateway technology, generating the in-frame fusion of Maltose Binding Protein (MBP) and MYB DNA-binding domain.

Transgenic *Arabidopsis* plants were generated using the floral dip method[49] and T1 transgenic plants were screened based on each selection marker of the construct. All oligonucleotides and synthesized fragment sequences used for this study are listed in Supplementary Table 3.

**Histochemical GUS staining**. Histochemical staining for GUS activity was performed by a common procedure[50] with some modifications. The thalli, antheridiophores, archegoniophores, and sporophytes of *pro*Mp*DUO1:GUS*-expressing plants were fixed in 90% (v/v) acetone, vacuum-infiltrated and incubated at 37 °C overnight in the GUS assay solution containing 100 mM sodium phosphate buffer (pH 7.2), 5 mM potassium-ferrocyanide, 5 mM potassium-ferricyanide, 0.1% (v/v) Triton X-100 and 0.5 mg ml⁻¹ 5-bromo-4-chloro-3-indolyl-β-D-glucuronic acid (X-Gluc). Chlorophyll in the tissue were removed by incubation in 70% (v/v) ethanol.

**RNA in situ hybridization**. Probe and tissue preparation and hybridization were performed as described[14]. A probe fragment for each gene was amplified from Tak-

1 cDNA with a set of gene-specific primers (Supplementary Table 3) and was cloned into the pCR-BluntII-TOPO vector (Life Technologies). DIG-labeled antisense and sense RNA probes were synthesized with a DIG RNA Labeling kit (SP6/T7) (Roche) according to the manufacturer's instructions. Antheridiophore receptacles of Tak-1 at stage 3 to 4[14] were fixed in a solution containing 3% (w/v) paraformaldehyde and 0.25% glutaraldehyde in 0.1 M phosphate buffer, pH 7.0, and 0.05% Triton X-100, dehydrated and embedded in paraffin. Eight-μm sections were made with a microtome, applied to an APS-coated glass slide, and then deparaffinized and rehydrated. They were treated with 1 μg ml⁻¹ of proteinase K (ThermoFisher Scientific) in 100 mM Tris-HCl, pH 8.0, 50 mM EDTA, pH 8.0 at 37 °C for 30 min, subsequently fixed in 4% (w/v) paraformaldehyde in PBS (7 mM Na₂HPO₄, 3 mM NaH₂PO₄, 130 mM NaCl) for 10 min, and treated with 0.5% (v/v) acetic anhydride in 100 mM triethanolamine for 10 min. Sections were incubated at 55 °C for 2 h in pre-hybridization buffer [50% formamide (w/v), 5 × SSC, 40 μg ml⁻¹ salmon sperm DNA] and then in the same buffer with 100 ng of probes for more than 16 h. After treatment with 50 μg ml⁻¹ of RNase A (SIGMA) in RNase buffer (10 mM Tris-HCl, pH 8.0, 500 mM NaCl, 1 mM EDTA) at 37 °C for 30 min, sections were washed in 0.2 × SSC (30 mM NaCl, 3 mM sodium citrate, pH 7.0) at 55 °C for 1 h, incubated with 1% (w/v) Blocking Reagent (Roche) in buffer 1 (100 mM Tris-HCl, pH 7.5, 150 mM NaCl) for 1 h and then incubated with 1/1000 diluted Anti-Digoxigenin-AP, Fab fragments (Roche) with buffer 1 containing 1% (w/v) blocking reagent for 1 h. The slides were subsequently washed three times with buffer 1 for 10 min each, rinsed with buffer 2 (100 mM Tris-HCl, pH 9.5, 100 mM NaCl, 50 mM MgCl₂) for 5 min and then covered with NBT/BCIP (Roche) diluted 1/125 in buffer 2. After incubation at 22 °C for more than 16 h in the dark, the reaction was stopped by immersing the slides in TE (10 mM Tris-HCl, 1 mM EDTA, pH 8.0).

**Confocal microscopy**. The accumulation of MpDUO1-Citrine was observed in isolated antheridia and sperm of Mp*DUO1-Citrine*[ki] plant. The Citrine fluorescence was detected in a range from 525 to 565 nm with confocal laser scanning microscopy (FV-1000; Olympus and LSM780; Zeiss) after excitation at 515 nm. An

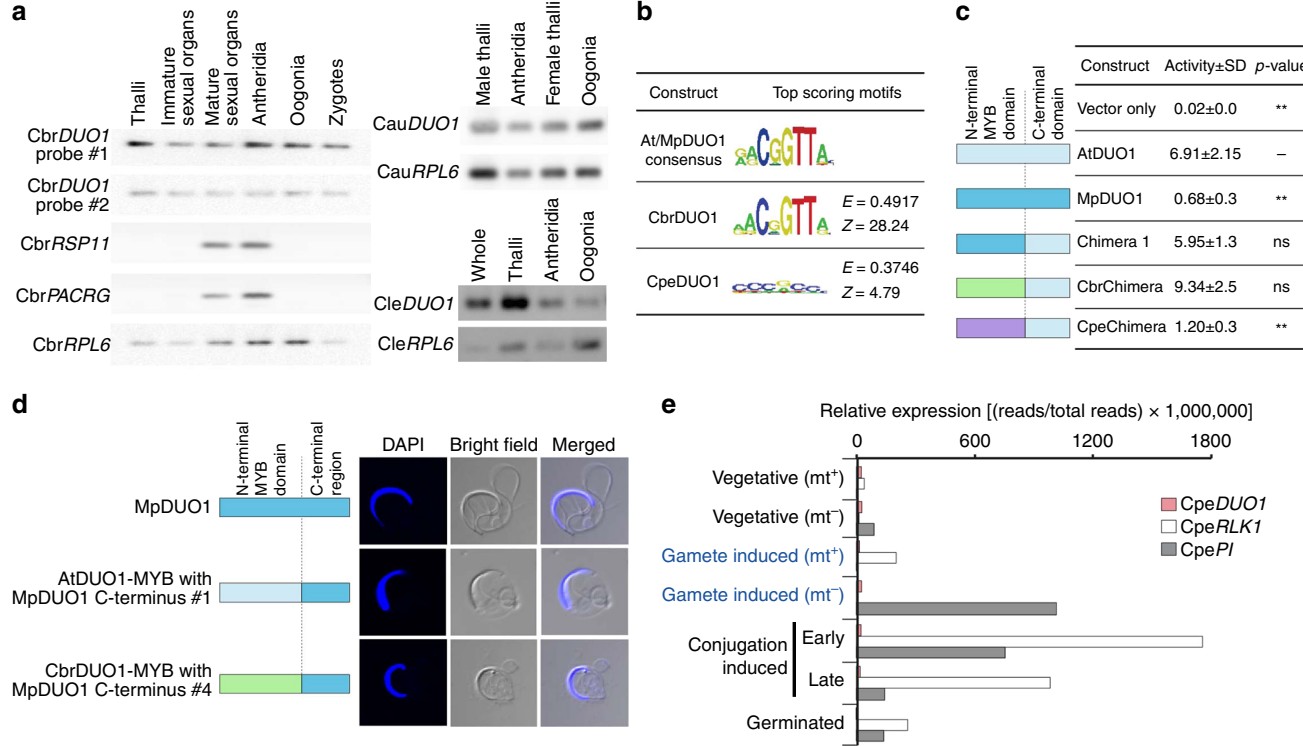

**Fig. 7** Characterization of green algal *DUO1* orthologs. **a** Expression profiles of *DUO1* orthologs from three *Chara* species, *C. braunii* (Cbr), *C. australis* (Cau), and *C. leptospora* (Cle) representing two distant clades (subgenera)[41]. Ribosomal protein L6 (*RPL6*) was used as the internal control. The expression of two flagella component genes (*RSP11* and *PACRG*) was also analyzed in *C. braunii*. Cbr*DUO1* gene structure and amplified probed regions are shown in Supplementary Figure 7a. The whole untrimmed images are shown in Supplementary Figure 7b. Oogonia contain the egg. Antheridia contain developing sperm. Thalli are somatic vegetative tissue. **b** Position weight matrix representations of the top-scoring 8-mer DNA sequences bound by different MYB domains on a protein-binding DNA microarray. Cpe*DUO1* is DUO1 from *Closterium peracerosum-strigosum-littorale* complex (conjugating green alga). Numbers denoted on the right side of each motif represent the motif E- and Z-scores[19]. **c** In vivo transcriptional activation potentials of green algal DUO1 chimeras. DUO1 transcriptional activation potentials were measured by relative luciferase activity (right) (*n* = 4; **p < 0.01; ns, not significant; Student's *t*-test). Schematic diagrams of constructs are color-coded light blue (AtDUO1), dark blue (MpDUO1), green (CbrDUO1), and purple (CpeDUO1). **d** The morphology of mature sperm of Mp*duo1-1*[ko] complemented by the expression of WT MpDUO1 (top), AtDUO1-MYB domain with MpDUO1 C-terminus chimera (middle), and CbrDUO1-MYB domain with MpDUO1 C-terminus chimera (bottom). The blue signals indicate the DAPI stained nuclei. Constructs are color-coded as in **c**. **e** Expression profiles of Cpe*DUO1* and sexual reproduction-specific Cpe*RLK1* and Cpe*PI* in *Closterium peracerosum-strigosum-littorale* complex. Stages equivalent to gametes in anisogamous plants are highlighted in blue. Relative expression ([reads per total sequenced reads] × 1,000,000) was used

antheridiophore of the Mp*DUO1-Citrine*[ki] plant was hand-sectioned with a blade and immersed in 1 µg ml$^{-1}$ 4′,6-diamidino-2-phenylindole (DAPI) solution. Antheridia were isolated from the sections and observed immediately under a confocal microscope (FV-1000, Olympus) with the following setting: MpDUO1-Citrine was excited by 488-nm laser and detected by a GaAsP detector with 535−565-nm window; DAPI was excited by 405-nm laser and detected by a photo-multiplier tube with 425−475-nm window.

Clover and mRuby2 in the transgenic *Arabidopsis* pollens were excited at 488 and 561 nm, respectively and detected in a range from 495 to 540 and from 566 to 600 nm, respectively, with confocal laser scanning microscopy (LSM780; Zeiss). In brief, 3-4 open flowers were collected in a microfuge tube containing 300 µl of the solution [0.1 M sodium phosphate (pH 7.0), 1 mM EDTA, 0.1% Triton X-100]. After brief vortexing and centrifugation, 15 µl of the pollen pellet was transferred to a microscope slide and imaged[12].

**Feulgen staining of antheridia.** Feulgen staining[51] was performed with some modifications. The antheridiophores receptacles of Tak-1 at stage 5[14] were fixed overnight in 3:1 mixture of ethanol and glacial acetic acid at 4 °C overnight. After fixation, the equal volume of ethanol was added and the samples were incubated for 1 h. Then the samples were hydrated in graded ethanol solutions. The samples were rinsed three times with distilled water for 15 min each and hydrolyzed in 5 N HCl for 1 h. After hydrolysis, the samples were rinsed three times with distilled water for 5 min each and stained with Schiff's reagent (Sigma-Aldrich) for 2 h. The samples were rinsed twice for 15 min each with distilled water and were dehydrated in graded ethanol solutions and 100% ethanol. The 100% ethanol was exchanged with the fresh one at hourly intervals until it remained colorless after the exchange. Then LR White resin (Sigma-Aldrich) was added to make a 1:1 mixture of 100% ethanol and LR White and the samples were left for 1 h at room temperature. Then

the mixture was replaced with pure LR White and left at room temperature overnight. The antheridia were manually dissected from antheridiophores and placed in fresh LR White on a standard glass microscope slide and a cover glass was gently lowered over the antheridia in LR White. The samples were incubated at 60 °C overnight. The cover glass can be carefully removed from the polymerized LR White. The fluorescence was detected at 535 nm and longer with LSM510 META Confocal Imaging System (Zeiss) after excitation at 488 nm by an argon laser.

**Nuclear shape quantification.** Fiji package[52] was used to quantify shapes [circularity ($4\pi$*area/perimeter$^2$), aspect ratio (major_axis/minor_axis), and solidity (area/convex_area)] of Feulgen-stained Marchantia antheridia nuclei obtained from Z-projected confocal images. In brief, nuclear images were smootened through Gaussian blur function, followed by threshold adjustment to capture the shape of nuclei. Nuclear images which are either partial or overlapping with others were excluded from quantification.

**TEM analysis of antheridia.** The antheridiophore receptacles of Tak-1 at stage 4[14] were fixed with 2% each of paraformaldehyde and glutaraldehyde in 0.05 M cacodylate buffer pH 7.4 at 4 °C overnight. After fixation, the samples were washed three times with 0.05 M cacodylate buffer for 30 min each, and were post-fixed with 2% osmium tetroxide in 0.05 M cacodylate buffer at 4 °C for 3 h. The samples were dehydrated in graded ethanol solutions then in 100% ethanol. The samples were infiltrated with propylene oxide twice for 30 min each and were put into a 7:3 mixture of propylene oxide and resin, Quetol-651 (Nissin EM Co.) for 1 h, then the mixture was kept in a tube without cap overnight to volatilize propylene oxide. The samples were transferred to a fresh resin and were polymerized at 60 °C for 48 h. The polymerized resins were ultra-thin sectioned at 80 nm with a diamond knife

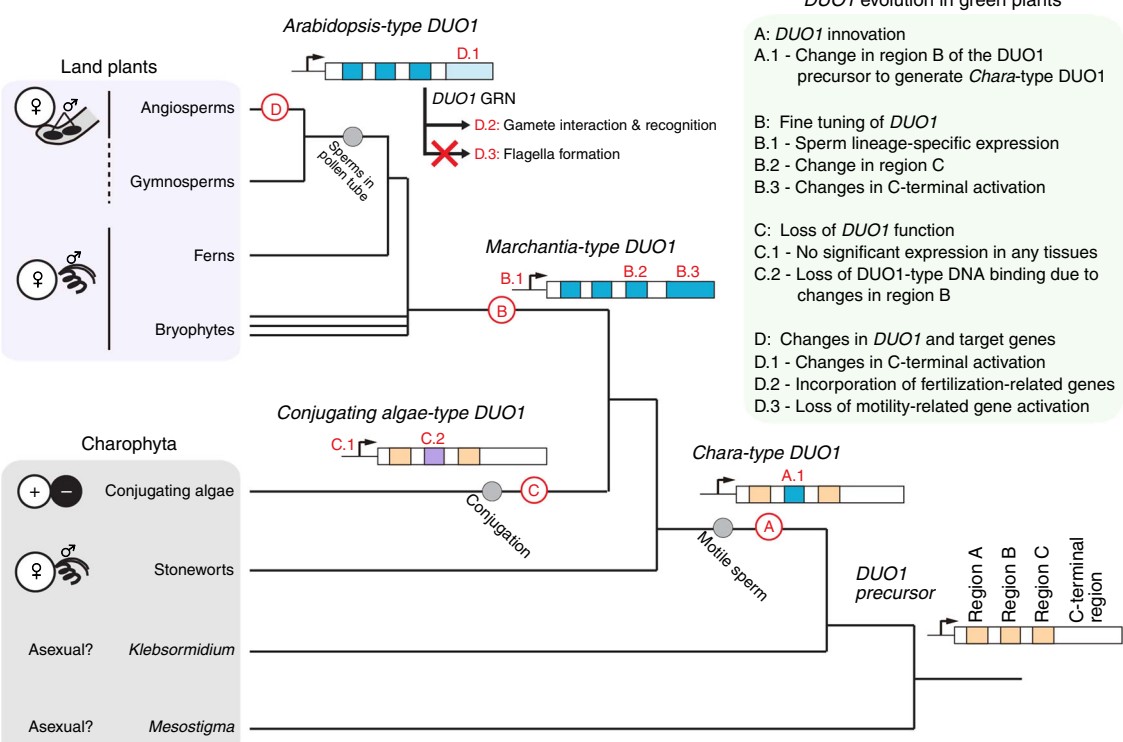

**Fig. 8** A model for *DUO1* evolution during green plant evolution. *DUO1* arose from a pre-existing MYB TF in an ancestor of stoneworts by acquiring a DUO1-type feature in region B of its MYB domain (A.1). *DUO1* innovation in ancestral plants promoted a specific program of differentiation of male gametes. Before the emergence of land plants, the sperm lineage-specific expression of *DUO1* was acquired (B.1) and a change in region C occurred to refine DNA-binding specificity (B.2), which has been conserved among land plants. The DUO1 C-terminal domain also underwent large changes, presumably adapting its transcription activation function for individual species during evolution (B.3). In conjugating green algae, the *DUO1* ortholog is no longer expressed in the sperm lineage (C.1) and has accumulated mutations in the MYB domain (C.2), resulting in a loss of DUO1 function. During land plant evolution, the DUO1 C-terminal activation domain has been modified, establishing lineage-specific activity (D.1). In seed plants, DUO1 retained its DNA-binding specificity and other male gamete-related genes, such as those important for fertilization, were recruited to (D.2) and motility-related genes were removed from (D.3) the set of genes controlled by DUO1, leading to significant rewiring of the *DUO1* regulatory network. Modes of sexual reproduction are shown on the left[37]. Sexual reproduction is unknown in *Mesostigma*, and *Klebsormidium*. Note that some gymnosperms (*Ginkgo* and cycads) have flagellate sperm cells delivered by a pollen tube

using Ultracut UCT (Leica) and the sections were mounted on copper grids. They were stained with 2% uranyl acetate at room temperature for 15 min, and then they were washed with distilled water followed by being secondary-stained with Lead stain solution (Sigma-Aldrich) at room temperature for 3 min. The grids were observed by JEM-1400Plus (JEOL Ltd.) at an acceleration voltage of 80 kV. Digital images were taken with a CCD camera, VELETA (Olympus).

**Immunofluorescence staining**. The fixation and permeabilization of antheridia were carried out essentially according to previous report[53]. Isolated spermatid cells were incubated with a polyclonal anti-centrin antibody against a recombinant protein corresponding to a full-length centrin from a brown alga *Scytosiphon lomentaria* (a gift from Dr. Taizo Motomura, Hokkaido University, Japan) diluted 1:500 with PBS [137 mM NaCl, 2.68 mM KCl, 8.1 mM Na$_2$HPO$_4$, and 1.47 mM KH$_2$PO$_4$, pH 7.4] in a moist chamber for 90 min at 37 °C. After a PBS wash, incubation with a monoclonal anti α-tubulin (T5168, Sigma) diluted 1:500 with PBS was performed for 60 min at 37 °C. After a wash again with PBS, the samples were incubated for 60 min at 37 °C with an equal mixture of Alexa 488-conjugated goat anti-rabbit IgG (H + L) (A11034, Invitrogen) diluted 1:200 with PBS and Alexa 568-conjugated goat anti-mouse IgG (H + L) (A11031, Invitrogen) diluted 1:200 with PBS. After washing with PBS for 10 min, the nuclei were stained with 1 µg ml$^{-1}$ DAPI in PBS. Slides were mounted using Vectashield (Vector Laboratory) and observed using a confocal laser scanning microscope (FV-1000, Olympus).

**Quantitative RT-PCR analysis**. Total RNA was extracted with an RNeasy Plant Mini Kit (QIAGEN) according to the manufacturer's protocol and the quality and quantity of the resultant RNA were evaluated using a NanoDrop 2000c spectrophotometer (ThermoFisher Scientific). One µg of total RNA was reverse-transcribed in a 20 µl reaction mixture using Transcriptor (Roche). After the reaction, the mixture was diluted with 180 µl of distilled water and 2 µl aliquots were used for PCR in a 10 µl PCR reaction mixture containing 1 µl of 10× Ex Taq

buffer, 1 µl of 2 mM dNTPs, 0.4 µl of 10 µM each of primers, and 0.05 µl of Ex Taq DNA polymerase (Takara) for semi-quantitative RT-PCR analysis. The PCR products were separated on a 2.5% (w/v) agarose gel, stained with ethidium bromide, and visualized under UV light. The primers used in these experiments are listed in Supplementary Table 3. For quantitative RT-PCR analysis, the cDNA samples were diluted with 220 µl of distilled water and 2 µl aliquots were amplified with the CFX96 Real-time PCR Detection System (Bio-Rad) using SYBR Premix Ex Taq (Tli RNaseH Plus) (Takara). The two-step PCR cycling program was performed according to manufacturer's protocol. The primers used in these experiments are listed in Supplementary Table 3. Mp*ACT1* was used as an internal control.

**Genetic complementation assay on At*duo1* and Mp*duo1-1*$^{ko}$**. For *Arabidopsis* complementation assay, mature pollen grains from WT and T3 stable homozygous complemented lines (AtComp, MpComp #3 and #4, and Chimera 1 #1 and #2) were examined by fluorescence microscopy. The frequency of bicellular and tri-cellular pollen grains was determined by scoring the number of respective pollen grains by DAPI staining. The ability of H3.10 activation was assessed by counting the frequency of pollen grains expressing mRuby2 fluorescence marker in sperm (*proAtHTR10:HTR10-mRuby2*).

For *Marchantia* complementation assay, the discharged sperm was directly observed without fixation on a Miniscope TM3000 (HITACHI, Japan) to obtain scanning electron microscopic images. To observe the nuclear morphology of sperm, the discharged sperm were stained with DAPI and observed under a confocal microscope (FV-1000, Olympus) with the following setting: DAPI was excited by 405-nm laser and detected by a photomultiplier tube with 425−475-nm window. Movies of discharged sperm were taken under a microscope BX43 (Olympus) with a dry dark-field condenser U-DCD (Olympus). For quantitative motility analysis, sperm discharged in water was observed under a phase-contrast/DIC microscope and video-recorded at the resolution of 1216 × 960 pixels and at the rate of 7 frames per second (fps) for 10 s by a microscope camera DP26 (Olympus). To track individual sperm, an open-source software ImageJ (ver. 1.51n,

the Fiji distribution)[52] was used. Movies of sperm were first converted into a sequence of individual frames. Their green-channel images were extracted and converted into 8-bit gray-scale, and their black and white values were inverted. To reduce stationary objects in the background, each image was subtracted with a Z projection of the entire frames. Moving paths of sperm were detected by TrackMate ver. 3.4.2[54], a plugin bundled with the Fiji distribution of ImageJ, with the following modified parameters: initial threshold, auto; linking max distance, 50 pixel; gap-closing max distance, 50 pixel. Tracks with duration of at least 3 s, or 21 frames, were selected and the length of each track was measured to calculate average speed in µm s$^{-1}$. Three movies were analyzed to obtain sperm swimming velocity for each genotype, and the total numbers of sperm observed are: $n = 132$ (WT), $n = 193$ (MpComp #5), $n = 14$ (AtComp #9), $n = 12$ (AtChimera #13), $n = 22$ (CbrChimera #1) for analysis of fraction of motile sperm; $n = 117$ (WT), $n = 67$ (MpComp #5), $n = 10$ (AtComp #9), $n = 8$ (AtChimera #13), $n = 12$ (CbrChimera #1) for analysis of swimming velocity.

To test the fertilization ability, mature antheridiophores were immersed in water and aliquots of sperm suspension were observed for discharged sperm and deposited onto archegoniophore receptacles of Tak-2. Sporophyte and sporangium development was observed about a month after crossing.

**Protein-binding DNA microarray assay.** Recombinant plasmids harboring MBP-MYB fusions were introduced into the BL21 strain of *E. coli*, and the expression of recombinant proteins was induced with 1 mM isopropyl β-D-1-thiogalactopyranoside (IPTG) for 6 h at 25 °C. Pellets corresponding to 25 ml of induced E. coli culture for each construct were stored at −80 °C and resuspended in 1 ml 1× binding buffer prior to DNA-binding assay[20]. Bacterial lysates were sonicated twice for 30 s, and centrifuged twice at 20,000×g to obtain cleared extracts of soluble proteins.

Second strand of DNA was synthesized in a primer extension reaction with 32 U Thermo Sequenase Polymerase (USB), 163 µM dNTPs, 1.63 µM Cy5-dUTP (GE Healthcare) and 1.17 µM oligonucleotide primer (5′-CAGCACGGACAACGGAA CACAGAC-3′) in a 900 µl total volume reaction. DNA microarray was incubated with the mixture in a hybridization oven for 10 min at 85 °C, the temperature gradually reduced up to 60 °C during 30 min and hold at this temperature for 90 min. The slide was then rapidly transferred to wash solution (1× PBS, 0.01% Triton X-100), incubated at 37 °C for 10 min with agitation and rinsed in 1× Phosphate Buffered Saline (PBS) for 3 min at room temperature. The slide was spun dry by centrifugation (1 min at 500 rpm) and scanned at 2 µm resolution in a Agilent's DNA Microarray Scanner for monitoring the amount of dsDNA. The binding mixture obtained from cleared bacterial lysates was adjusted to 175 µl and to contain 2% milk and 0.89 µg of denatured salmon sperm DNA (ssDNA). Double stranded DNA microarray was incubated with the binding mixture in a humid chamber for 2.5 h at room temperature. Slides were then washed three times PBS-1% Tween 20 (5 min), three times in PBS-0.01% Triton X-100 (5 min) and spun dry by centrifugation. DNA–protein complexes were incubated with 16 µg of Rabbit polyclonal to Maltose Binding Protein (Abcam) in PBS-2% milk for 16 h at room temperature. Slides were washed 3× in PBS-0.05% Tween 20, 3× in PBS-0.01% Triton X-100 (5 min each wash) and dried. Labeling of DNA–protein complexes were performed by incubating the microarrays with 0.4 µg of goat anti-rabbit IgG DyLight 549 conjugated (Pierce) in PBS-2% milk for 3 h at room temperature, followed by the same washes as before and the slides dried for scanning.

We used the nPBM11 design containing 167,773 different oligonucleotide probes[19] synthesized in an Agilent's SurePrint G3 4 × 180k format (Agilent Technologies). DNA microarrays were scanned in a DNA Microarray Scanner at 2-µm resolution and quantified with Feature Extraction 9.0 software (Agilent Technologies). Normalization of probe intensities and calculation of E- and Z-scores of all the possible 8-mers were carried out with the PBM Analysis Suite[55]. Perl scripts were modified to adapt them to nPBM11 microarray dimensions and to input files generated by Feature Extraction software.

**Transient luciferase assay.** *Agrobacterium*-mediated transient transformation of *Nicotiana tabacum* leaf was carried out as described by Sparkes et al.[56] modified as detailed below. *Agrobacterium* strains were combined as required at an OD600 of 0.1 for reporter and effector vectors and an OD600 of 0.02 for the *Renilla* luciferase control vector in infiltration media (280 mM D-glucose, 50 mM MES, 2 mM Na$_2$PO$_4$·12H$_2$O, 0.1 mM acetosyringone). 4–6-week-old *Nicotiana tabacum* plants were grown in greenhouse conditions. *Agrobacterium* suspensions were taken up in 1 ml syringes and the underside of leaves gently rubbed to remove a small region of the cuticle. The syringe tip was placed at these regions and *Agrobacterium* suspensions were gently infiltrated. Plants were placed in a growth chamber under normal growth conditions and left for 2 days. Each leaf disc from an infiltrated region was excised using a 9 mm cork-borer and ground to homogeneity in 300 µl of 1× Passive Lysis Buffer (Promega) in a chilled mortar and pestle. Leaf extracts were centrifuged at 16,000×g for 5 minutes at 4 °C to pellet cell debris. Two separate 25 µl aliquots were assayed separately for firefly and *Renilla* luciferase activities in 100 µl of the respective assay buffer. The firefly luciferase assay buffer (25 mM glycylglycine, 15 mM KPO$_4$ pH 8.0, 4 mM EGTA, 2 mM ATP, 1 mM DTT, 15 mM MgSO$_4$, 0.1 mM CoA, 75 µM luciferin with final pH adjusted to 8.0) and *Renilla* luciferase assay buffer (1.1 M NaCl, 2.2 mM Na$_2$EDTA, 0.22 M KPO$_4$ pH

5.1, 0.44 mg/ml BSA, 1.43 µM coelenterazine with final pH adjusted to 5.0) were prepared immediately before measurement[57]. Luminescence was measured in white 96-well plates with a FLUOstar Omega (BMG LABTECH Ltd) microplate reader as relative luminescence units (RLUs) integrated over 10 seconds. Extracts of discs taken from non-infiltrated leaves were assayed to determine mean background RLU values, which were subtracted from those for extracts of infiltrated leaves. Normalized luciferase activity (FLuc/RLuc) was calculated for each extract from the ratio of background subtracted RLUs obtained in firefly luciferase (FLuc) and *Renilla* (RLuc) luciferase assays.

**Sequences, alignment, and phylogenetic tree construction.** Sequences of *Klebsormidium* MYB TFs and DUO1 MYB TFs of conjugating green algae were obtained from published data[58–60]. The 5′ and 3′ parts of Cpe*DUO1* cDNA fragment were amplified separately by PCR from cDNA from conjugation induced early stage. Sequences of land plant *DUO1, DAZ1, GCS1/HAP2*, and *GEX2* were obtained from www.phytozome.jgi.doe.gov. Amino acid sequence alignment illustrated in Supplementary Figure 1a was generated by CLC Workbench 7 package (QIAGEN).

**RNA preparation and RNA-seq analysis.** For the preparation of total RNA from vegetative cultures of *Closterium peracerosum-strigosum-littorale* complex, mating-type (mt)$^+$ and mt$^-$ cells were harvested at 0 (start time of light illumination), 6, 12, and 18 h, respectively. For the preparation of total RNA from mating cultures, vegetative growing cells of the mt$^+$ and mt$^-$ were collected, washed three times with nitrogen-depleted medium (MI medium)[61], and incubated separately in MI medium (3.6 × 10$^5$ cells/72 ml in 300-ml Erlenmeyer flasks) under continuous light for 2, 8, and 24 h (gamete induced mt$^+$ and mt$^-$, respectively). The cells of both mating types, which had been separately cultured in MI medium (at 3.6×10$^5$/72 ml in 300-ml Erlenmeyer flasks) for 24 h, were mixed and co-incubated (at 3.6 × 10$^5$ each/72 ml in 300-ml Erlenmeyer flasks) for 1, 2, 4, 6, 8, 12, 16, 20, and 24 h (conjugation induced early) and for 48, 72, 96 h (conjugation induced late). For the preparation of total RNA from germinated zygote, zygotes were dried once and incubated in C medium for 12, 24, 48, 72 h under a 16 h light/8 h dark cycle (germinated). The harvested cells were frozen in liquid nitrogen and total RNA was isolated using TRIZOL Plus Kit (Invitrogen, Carlsbad, CA, USA), in accordance with supplier instructions. Paired-end libraries were generated with TruSeq RNA Sample Preparation Kit (Illumina, www.illumina.com), according to manufacturer's instructions. Sequencing was carried out 76 bps with a Genome Analyzer IIx using standard reagents. All high-quality sequences were de novo assembled with Trinity[62]. Expression frequency of the contigs was calculated by RSEM[63], using all high-quality sequences.

**DUO1 DNA-binding motif search.** The DUO1 consensus DNA-binding motif (5′-RRCSGTT-3′) generated in this study was used to search the upstream regions of At*DAZ1*, At*GCS1/HAP2*, and At*GEX2* homologs of some angiosperms, a lyco-phyte, and some bryophytes by using the dna-pattern tool from Regulatory Sequence Analysis Tools (RSAT) web server (http://www.rsat.eu/)[64] with default parameters. Up to 2-kb upstream regions between the putative transcriptional start site and the upstream neighboring gene were collected from Phytozome ver.11 (https://phytozome.jgi.doe.gov/pz/portal.html). The genes used in this experiment are listed in Supplementary Table 4.

**Selection pressure analyses.** Branch model tests were performed using codeML in PAML[65] on DUO1 and S18 MYB domain sequences from genes in Supplementary Table 4[15,58,59]. Default parameters were used, to the exception of clean-data = 0 to prevent removal of codon information around region B. Likelihood ratio tests were performed between models and chi-squared tests were performed to assess statistical significance. Maximum-likelihood tree shown in Supplementary Figure 4 was generated in PhyML 3.1[66] using the LG + I + G protein substitution model, selected according to ProtTest 3[67]. Full results are listed in Supplementary Table 2.

**RT-PCR and Southern hybridization analysis.** Gene expression analysis of three *Chara* species was performed using standard procedures[40] with the following modifications. A probe fragment for each gene was amplified from cDNA of each species with a set of gene-specific primers (Supplementary Table 3) and was cloned into the pCR-BluntII-TOPO vector (Life Technologies), which contains SP6 and T7 polymerase-binding sites. RT-PCR was performed with a set of gene-specific primers (Supplementary Table 3). Probe synthesis, hybridization, washing, and detection were performed by using DIG High Prime DNA Labeling and Detection Starter Kit II (Roche) according to the manufacturer's protocol. Hybridization was performed at 55 °C.

## Data availability
Novel data generated in this study have been deposited at GenBank under the accessions: MpDUO1 (LC172177), MpR2R3-MYB07 (KX683859), MpR2R3-MYB21 (KX683860) MpACT1 (LC172182) MpCEN1 (LC379265) MpTUA5 (LC172181) MpDAZ1 (LC172178) MpGCS1 (LC172179) MpGEX2 (LC172180)

HmnDUO1 (LC379264) HmnACT1 (LC379378) CbrDUO1 (LC199499), Cau-DUO1 (LC221833), CleDUO1 (LC221832), CbrRSP11 (LC382020) CbrPACRG (LC382019) CbrRPL6 (LC382018) CauRPL6 (LC382017) CpeDUO1 (LC176570). Previously reported sequences used in this study are available at GenBank under the accession: MpPACRG (LC102460), MpPRM (LC102462) MpTUB4 (KM096548) MpLC7 (LC102461), MpMID/RWP2 (KU987912) CleRPL6 (AB035569), CpeRLK1 (AB920609) CpePI (AB012698). All other data are available from the authors on reasonable request.

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

## Acknowledgements
We thank M. Endo, S. Akimcheva, B. Jamge, K. Nagao, and L. Collins for technical assistance and suggestions and Dominique Bergmann for critical reading of the manuscript. We thank T. Motomura, C. Nagasato, K. Ura (Hokkaido University, Japan), and K. Kimura (Saga University, Japan) for the anti-centrin antibody, and the Vienna Biocenter Core Facilities for technical help. This work was supported by FWF (I2163-B16 to F.B. and W1238-B20 to S.A.M.) and the Biotechnology and Biological Research Council (BB/N005090 to D.T.); ERA-CAPS EVOREPRO project to DT and FB; Spanish MINECO grant BIO2017-86651-P (AEI/FEDER) to J.F.-Z.; JSPS KAKENHI grant 15K07185 to H. Sa, MEXT grants (25113005, 23370022, and 24657031 to T.A.; 25113001 and 15K21758 to T.Ko. and T.A.; and 221S0002 to H. Sa., H. Se., and T.N.; and 26291081 to T.N., KS, and M.S.); a Grant-in-Aid for the Japan Society for the Promotion of Science Fellows (to A.H. and A.O.); the Kyoto University BRIDGE program (to A.H.); Lise-Meitner fellowship (M1818-B21 to M.B.); and GMI (T.Ka., M.B., O.A., and F.B.).

## Author contributions
A.H., T.Ka., F.B., T.A., and D.T. conceived the study; A.H., T.Ka., M.B., M.Z., D.H., M.S., T.A., Y.T., T.T., K.K., and A.O. performed molecular and cellular analyses; I.L.-V. and J. F.-Z carried out DNA microarray experiments; S.M. performed molecular evolution analysis; H.Se., H.Sa., T.N., and Y.S. performed algal genome and expression analyses; K. Y., K.I., R.N., and T.Ko. generated *Marchantia* genome resources and vectors; T.N., KS, and M.S. identified Hmn*DUO1*; A.H., T.Ka., J.F.-Z., D.T., F.B., and T.A. analyzed the data; T.Ka., F.B., and A.H. wrote the manuscript with critical reviewing and editing by T. A., D.T., M.B., and J.F.-Z.

## Additional information

**Competing interests:** The authors declare no competing interests.

