## [Peer Review File · Nature Communications]

Reviewer #1 (Remarks to the Author):

This 25-authored paper concerns how sperm cell differentiation is conducted in plants and focuses principally on the R2R3-MYB transcription factor DUO POLLEN1, or as more commonly abbreviated, DUO1, and its evolution, as examined by looking at the function of DUO1 and DUO1-controlled genes that are expressed in principally angiosperms (most prevalent, AtDUO1) and in early land plants, principally represented by Mp (Marchantia polymorpha), in MpDUO1. AtDUO1 controls angiosperm male gametes by licensing the precursor generative cell to undergo DNA synthesis to be able to proceed to G2 and allow mitosis of the generative to undergo mitosis to form two sperm cells, followed by cellular differentiation of these product cells into a sperm cell fate. *duo1* null mutant cells do not undergo G2 to M transition in the generative cell and sperm cells are not formed and do not mature. MpDUO1 regulates sperm differentiation in earlier land plants (represented by Mp) controlling the maturation of characteristic sperm cell features. These sperm cell features include differentiation of the typical sperm cytoskeleton through which tubulin gene are activated to display motility, as well as other critical components of the locomotory function, causing e.g., failure of axoneme organization, failure of a dynein light chain activation and absence of a required protamine-like arginine-rich protein associated with nuclear condensation in sperm--all features that are critical to function.

There is also a conserved DUO1-controlled interaction with DAZ1, which is a downstream transcription factor of Mp. In Mp, this inhibits sperm-critical features from developing and impedes function. In At, AtDUO1 interacts with AtGCS1, which controls gamete attachment and separately interacts with GEX2, which is involved with sperm cell adhesion. Thus, the earlier land plants fail to function because of poor or insufficient motility, whereas in angiosperms, there are gamete fusion-related defects. Interestingly, AtDUO1 also drives histone H3.10 substitution of an H3 histone. Similar H3 histone substitution is not found in Mp. It would be interesting if there is a general impoverishment of such histone modification. Earlier algal species lack the requirement to produce DUO1 and thus this transcription factor is not found except in land plants. Thus it is reserved to land plants.

Interestingly there is essentially no discussion of the broader significance of the MYB family, such as even citation of Du H, Liang Z, Zhao S, Nan MG, Tran LS, Lu K, Huang YB, Li JN. The evolutionary history of R2R3-MYB proteins across 50 eukaryotes: new insights into subfamily classification and expansion. *Nature Scientific Reports*. 2015 Jun 5;5:11037. Ironically, the myopia spreads in both directions as they fail to cite the importance of MYB genes (e.g., the specific MYB gene DUO1) with critical importance in licensing sperm cells, without which there would be no sexual reproduction in land plants.

Algal precursors have MYBs but none having the unique features of the DUO1 related genes, including lacking the conservation of the gene regions that determine binding specificity. For

example, the exchange of region B or C reduced their trans-activation potential. Thus, this motif is a DUO1-determining characteristic that confers characteristic function to the gene.

Most importantly a number of sperm specific characteristics are regulated by MpDUO1 in the sperm cell. It could be argued that *Marchantia polymorpha* may not be characteristic of early land plants, but they appear to conserve the defining, critical genes that land plants possess and appear to incorporate neofunctionalization that became further refined during evolution.

Reviewer #2 (Remarks to the Author):

In their manuscript "Molecular origin of sperm differentiation of plants" Higo et al. report the evolution of DUO1-type MYB TFs and their role for sperm differentiation in the land plant lineage. Their report includes expression studies of various DUO1 homologs, generation and complementation of mutants, transactivation and DNA-protein interaction studies as well as the application of phylogeny and modelling. It has been reported before that DUO1 is essential for male germ cell specificity in *Arabidopsis*, but this study essentially expands our knowledge about DUO1 in the whole plant lineage and additionally demonstrates how the evolution of transcriptional regulators can be studied in general.

I am very enthusiastic about the report and have only a few suggestions to further improve the manuscript.

In general, I agree with the authors that DUO1 is a key for the emergence and maintenance of sperm differentiation. However, there might be other key regulators and I thus find the title a bit overdone indicating that the authors have found the holy grail of sperm differentiation in plants - I think they describe a (perhaps the) major regulator and I suggest they name DUO1 in the title.

I also think that the model shown in Suppl. Fig. 6 is a very important summary of the partially very complex findings shown in the manuscript: it would benefit if the figure would be moved into the main manuscript.

MINOR

Figure 1a is not properly referred to in the text; thus it is either superfluous or the authors modify the text accordingly; in the legend “A” should be written with small lettering;

Fig. 1a: if the image is shown, the mode of sperm movement should be indicated for all plant branches;

Fig. 2a: I find it difficult to judge the presence of sperm from the images – perhaps the authors could show sperm mass in a different way?

Fig. 2f: the statistics of the chimera combining AtDUO1 DNA binding domain with the MpDUO1 C-terminal activation domain should also be included in the graph;

Fig. 3a: I am surprised that S should interact with the DNA and not the residues of conserved K- and R-amino acids; I also think that a movie of the model is more helpful than the movies attached showing sperm cells;

Fig. 3b: how do they explain the significant different expression strength of Chimera 1 in the top and bottom experiment?

Fig. 3c: Chimera 3 is missing;

Line 13 p : “We wished to study ...” sounds strange - this has been done; “Next we studied ...” or similar is better suited to introduce the paragraph.

Reviewer #3 (Remarks to the Author):

Molecular origin of sperm differentiation in plants

Higo et al.

The authors investigate the role of DUO in specifying sperm cell fate throughout land plants. They demonstrate that DUO arose in the charophycean algae by mutations in the middle region of the DNA binding domain that conferred an altered DNA binding capability. While the change occurred in the common ancestor of Chara + land plants, the restriction of DUO's function to sperm cells did not occur until the ancestral land plant. Remarkably, the regulatory elements controlling DUO's sperm cell specific expression is also conserved across land plants. The conclusions are supported by the results presented.

In the final version the figures should be larger — it is difficult to look at some of the panels without the aid of a magnifying glass.

Minor comments:

Page 3, line 58: can it really be called sperm in the green algae? — or merely a gamete of a particular mating type?

Page 12, line 184: might just state that it is broadly expressed, and not restrictive to the sperm lineage

Page 12: With so much data presented, it seems presumptuous to ask for more, but was a DUO ortholog found in the Coleochaetales? Many of these taxa also have motile sperm and diverged after the Charales, and thus analysis in these taxa would pinpoint the time in evolution where DUO became sperm specific.

Response to REVIEWERS' COMMENTS:

Reviewer #1 (Remarks to the Author):

This 25-authored paper concerns how sperm cell differentiation is conducted in plants and focuses principally on the R2R3-MYB transcription factor DUO POLLEN1, or as more commonly abbreviated, DUO1, and its evolution, as examined by looking at the function of DUO1 and DUO1-controlled genes that are expressed in principally angiosperms (most prevalent, AtDUO1) and in early land plants, principally represented by Mp (Marchantia polymorpha), in MpDUO1. AtDUO1 controls angiosperm male gametes by licensing the precursor generative cell to undergo DNA synthesis to be able to proceed to G2 and allow mitosis of the generative to undergo mitosis to form two sperm cells, followed by cellular differentiation of these product cells into a sperm cell fate. duo1 null mutant cells do not undergo G2 to M transition in the generative cell and sperm cells are not formed and do not mature. MpDUO1 regulates sperm differentiation in earlier land plants (represented by Mp) controlling the maturation of characteristic sperm cell features. These sperm cell features include differentiation of the typical sperm cytoskeleton through which tubulin gene are activated to display motility, as well as other critical components of the locomotory function, causing e.g., failure of axoneme organization, failure of a dynein light chain activation and absence of a required protamine-like arginine-rich protein associated with nuclear condensation in sperm--all features that are critical to function.

There is also a conserved DUO1-controlled interaction with DAZ1, which is a downstream transcription factor of Mp. In Mp, this inhibits sperm-critical features from developing and impedes function. In At, AtDUO1 interacts with AtGCS1, which controls gamete attachment and separately interacts with GEX2, which is involved with sperm cell adhesion. Thus, the earlier land plants fail to function because of poor or insufficient motility, whereas in angiosperms, there are gamete fusion-related defects. Interestingly, AtDUO1 also drives histone H3.10 substitution of an H3 histone. Similar H3 histone substitution is not found in Mp. It would be interesting if there is a general impoverishment of such histone modification. Earlier algal species lack the requirement to produce DUO1 and thus this transcription factor is not found except in land plants. Thus it is reserved to land plants.

Interestingly there is essentially no discussion of the broader significance of the MYB family, such as even citation of Du H, Liang Z, Zhao S, Nan MG, Tran LS, Lu K, Huang YB, Li JN. The evolutionary history of R2R3-MYB proteins across 50 eukaryotes: new insights into subfamily classification and expansion. Nature Scientific Reports. 2015 Jun 5;5:11037. Ironically, the myopia spreads in both directions as they fail to cite the importance of MYB genes (e.g., the specific MYB gene DUO1) with critical importance in licensing sperm cells, without which there would be no sexual reproduction in land plants.

Reply: We now cite this reference in the text “DUO1-type MYB TF is present in land plants¹⁵

Algal precursors have MYBs but none having the unique features of the DUO1 related genes, including lacking the conservation of the gene regions that determine binding specificity. For example, the exchange of region B or C reduced their trans-activation potential. Thus, this motif is a DUO1-determining characteristic that confers characteristic function to the gene. Most importantly a number of sperm specific characteristics are regulated by MpDUO1 in the sperm cell. It could be argued that Marchantia polymorpha may not be characteristic of early

land plants, but they appear to conserve the defining, critical genes that land plants possess and appear to incorporate neofunctionalization that became further refined during evolution.

Reply: We agree with Reviewer#1 that without rich, well-preserved fossils representing earliest land plants and well-resolved molecular phylogeny to decisively conclude which of the three bryophyte groups is closest to the earliest land plants, no one can be sure whether *Marchantia polymorpha* is or is not characteristic of earliest land plants in terms of plant body morphology. However, as to the sperm morphology, we can safely argue that it likely represents the state of the earliest land plants based on fairly similar sperm among liverworts and Charophytes that represent a much ancient ancestor common with bryophytes.

Reviewer #2 (Remarks to the Author):

In their manuscript “Molecular origin of sperm differentiation of plants” Higo et al. report the evolution of DUO1-type MYB TFs and their role for sperm differentiation in the land plant lineage. Their report includes expression studies of various DUO1 homologs, generation and complementation of mutants, transactivation and DNA-protein interaction studies as well as the application of phylogeny and modelling. It has been reported before that DUO1 is essential for male germ cell specificity in Arabidopsis, but this study essentially expands our knowledge about DUO1 in the whole plant lineage and additionally demonstrates how the evolution of transcriptional regulators can be studied in general.

I am very enthusiastic about the report and have only a few suggestions to further improve the manuscript.

In general, I agree with the authors that DUO1 is a key for the emergence and maintenance of sperm differentiation. However, there might be other key regulators and I thus find the title a bit overdone indicating that the authors have found the holy grail of sperm differentiation in plants - I think they describe a (perhaps the) major regulator and I suggest they name DUO1 in the title.

Reply: we agree and provide a new title: “Transcription factor DUO1 generated by neofunctionalization was associated with evolution of sperm differentiation in plants”

I also think that the model shown in Suppl. Fig. 6 is a very important summary of the partially very complex findings shown in the manuscript: it would benefit if the figure would be moved into the main manuscript.

Reply : We agree and have now include the model in the main text as Figure 8.

MINOR

Figure 1a is not properly referred to in the text; thus it is either superfluous or the authors modify the text accordingly; in the legend “A” should be written with small lettering;

Reply : We agree and have corrected this

Fig. 1a: if the image is shown, the mode of sperm movement should be indicated for all plant branches;

Reply : We agree and have added this information

Fig. 2a: I find it difficult to judge the presence of sperm from the images – perhaps the authors could show sperm mass in a different way?

Reply: sperm masses are shown by the white aggregates in WT whereas there is no such aggregates in the mutant. This is highlighted now in the text and images were contrasted a bit more to enhance visualisation.” Production of sperm as white aggregates is observed from WT antheridiophores but not in Mpduo1^{ko} (Fig. 2a, b)._

Fig. 2f: the statistics of the chimera combining AtDUO1 DNA binding domain with the MpDUO1 C-terminal activation domain should also be included in the graph;

Reply : We agree and have added this information in what is now Fig3a

Fig. 3a: I am surprised that S should interact with the DNA and not the residues of conserved K- and R-amino acids; I also think that a movie of the model is more helpful than the movies attached showing sperm cells;

Reply : We are sorry but are unable to produce a movie from the model

Fig. 3b: how do they explain the significant different expression strength of Chimera 1 in the top and bottom experiment?

Reply: The strength varies because the experiments were done several apart on plants grown in different conditions and this is why controls need to be repeated for each experimental set.

Fig. 3c: Chimera 3 is missing;

Reply: Chimera 3 was not tested because we knew already from the tests in planta that its activity does not differ significantly from chimera1

Line 13 p : “We wished to study ...” sounds strange - this has been done; “Next we studied ...” or similar is better suited to introduce the paragraph.

Reply : We agree and have corrected this

Reviewer #3 (Remarks to the Author):

Molecular origin of sperm differentiation in plants
Higo et al.

The authors investigate the role of DUO in specifying sperm cell fate throughout land plants. They demonstrate that DUO arose in the charophycean algae by mutations in the middle

region of the DNA binding domain that conferred an altered DNA binding capability. While the change occurred in the common ancestor of Chara + land plants, the restriction of DUO's function to sperm cells did not occur until the ancestral land plant. Remarkably, the regulatory elements controlling DUO's sperm cell specific expression is also conserved across land plants. The conclusions are supported by the results presented.

In the final version the figures should be larger — it is difficult to look at some of the panels without the aid of a magnifying glass.

Minor comments:

Page 3, line 58: can it really be called sperm in the green algae? — or merely a gamete of a particular mating type?

Reply : We agree and have corrected this to male gamete identity

Page 12, line 184: might just state that it is broadly expressed, and not restrictive to the sperm lineage

Reply : We agree and have corrected this to “they express a DUO1 ortholog (Fig. 1b) with a pattern broader than the sperm specific DUO1 expression in land plants”

Page 12: With so much data presented, it seems presumptuous to ask for more, but was a DUO ortholog found in the Coleochaetales? Many of these taxa also have motile sperm and diverged after the Charales, and thus analysis in these taxa would pinpoint the time in evolution where DUO became sperm specific.

Reply : We agree on principle but Coleochaetales genomes have not been fully sequenced and to our knowledge isolation of gamete was not successful in any species from this group and as a result we have not been able to identify an ortholog of Duo1 from Coleochaetales. We issue a statement in the text “It will be of particular interest to know whether an ortholog of DUO1 exists in Coleochate, which belongs to Charophyte and produces motile bi-flagellate sperm similar to those of liverworts³⁷. But this lack of identification may be the result of limited availability of genome and transcriptome to vegetative cells only.”